# Optogenetic dissection of mitotic spindle positioning in vivo

**Lars-Eric Fielmich[1†], Ruben Schmidt[1,2†], Daniel J Dickinson[3,4‡], Bob Goldstein[3,4], Anna Akhmanova[2], Sander van den Heuvel[1]***

[1]Developmental Biology, Department of Biology, Faculty of Sciences, Utrecht University, Utrecht, Netherlands; [2]Cell Biology, Department of Biology, Faculty of Sciences, Utrecht University, Utrecht, Netherlands; [3]Lineberger Comprehensive Cancer Center, University of North Carolina at Chapel Hill, Chapel Hill, United States; [4]Department of Biology, University of North Carolina at Chapel Hill, Chapel Hill, United States

***For correspondence:**
s.j.l.vandenheuvel@uu.nl

[†]These authors contributed equally to this work

**Present address:** [‡]Department of Molecular Biosciences, University of Texas at Austin, Austin, United States

**Abstract** The position of the mitotic spindle determines the plane of cell cleavage, and thereby daughter cell location, size, and content. Spindle positioning is driven by dynein-mediated pulling forces exerted on astral microtubules, which requires an evolutionarily conserved complex of Gα•GDP, GPR-1/2[Pins/LGN], and LIN-5[Mud/NuMA] proteins. To examine individual functions of the complex components, we developed a genetic strategy for light-controlled localization of endogenous proteins in *C. elegans* embryos. By replacing Gα and GPR-1/2 with a light-inducible membrane anchor, we demonstrate that Gα•GDP, Gα•GTP, and GPR-1/2 are not required for pulling-force generation. In the absence of Gα and GPR-1/2, cortical recruitment of LIN-5, but not dynein itself, induced high pulling forces. The light-controlled localization of LIN-5 overruled normal cell-cycle and polarity regulation and provided experimental control over the spindle and cell-cleavage plane. Our results define Gα•GDP–GPR-1/2[Pins/LGN] as a regulatable membrane anchor, and LIN-5[Mud/NuMA] as a potent activator of dynein-dependent spindle-positioning forces.
DOI: https://doi.org/10.7554/eLife.38198.001

## Introduction

Animal cells control the position of the spindle to determine the plane of cell cleavage. Regulated spindle positioning is therefore critical for asymmetric cell division and tissue formation (*di Pietro et al., 2016*). Early work in *C. elegans* demonstrated that cortical pulling forces position the spindle through a protein complex that consists of a heterotrimeric G protein alpha subunit, GOA-1[Gαo] or GPA-16[Gαi] (together referred to as Gα), a TPR-GoLoco domain protein GPR-1/2, and the coiled-coil protein LIN-5 (*Colombo et al., 2003*; *Gotta and Ahringer, 2001*; *Gotta et al., 2003*; *Grill et al., 2001*; *Lorson et al., 2000*; *Miller and Rand, 2000*; *Srinivasan et al., 2003*). This complex, and the closely related *Drosophila* Gα[i/o]–Pins–Mud and mammalian Gα[i/o]–LGN–NuMA protein complexes, recruit the microtubule motor dynein to the cell cortex (*Bellaïche et al., 2001*; *Bowman et al., 2006*; *Du and Macara, 2004*; *Du et al., 2001*; *Izumi et al., 2004*; *Nguyen-Ngoc et al., 2007*; *Schaefer et al., 2001*; *Zheng et al., 2010*; *Zhu et al., 2011*) (*Figure 1a*). While regulation at the level of individual components has been described, it remains unclear whether these proteins only form a physical anchor for dynein, or whether individual subunits contribute additional functions in spindle positioning.

As a potential additional function, force generation may require a dynein adaptor that activates dynein motility. Such an adaptor is necessary for the processive movement of mammalian cytoplasmic dynein during cargo transport along microtubules (*Reck-Peterson et al., 2018*). This process differs substantially from microtubule-dependent cortical pulling, in which force is generated by

---

**eLife digest** A cell about to divide must decide where exactly to cut itself in two. Split right down the middle, and the two daughter cells will be identical; offset the cleavage plane to one side, and the resulting siblings will have different sizes, places and fates.

In animals, the splitting of cells is dictated by the location of the spindle, a structure that forms when cable-like microtubules stretch from the cell membrane to attach to the chromosomes. At the membrane, a group of proteins tugs on the microtubules to bring the spindle into the correct position. One of these proteins, dynein, is a motor that uses microtubules as its track to pull the spindle into place. What the other parts of the complex do is still unclear, but a general assumption is that they may be serving as an anchor for dynein.

To test this model, Fielmich, Schmidt et al. removed one or more proteins from the complex in the developing embryos of the nematode worm *Caenorhabditis elegans*. A light-activated system then linked the remaining proteins to the membrane by tying them to an artificial anchor. Two of the proteins in the complex could be replaced with the artificial anchor, but pulling forces were absent when dynein was artificially tied to the membrane. This indicates that the motor being anchored at the edge of the cell is not enough for it to pull on microtubules. Instead, the experiments showed that dynein needs to be activated by another component of the complex, a protein called LIN-5. This suggests that individual proteins in the complex have specialized roles that go beyond simply tethering dynein. In fact, steering where LIN-5 was attached on the membrane helped to control the location of the spindle, and therefore of the cleavage plane.

As mammals have a protein similar to LIN-5, dissecting the roles of the components involved in positioning the spindle in *C. elegans* could help to understand normal and abnormal human development. In addition, these results demonstrate that creating artificial interactions between proteins using light is a powerful technique to study biological processes.

DOI: https://doi.org/10.7554/eLife.38198.002

---

dynein in association with shrinking microtubules (*Laan et al., 2012*). Without adaptor, surface-anchored yeast dynein in contact with depolymerizing microtubules generates pulling forces in vitro (*Laan et al., 2012*). However, yeast dynein moves processively on its own (*Reck-Peterson et al., 2006*). Hence, it remains unknown whether cortical pulling force generation in animal cells depends just on anchoring of dynein, or whether this requires an additional dynein activator.

The role of $G\alpha_{i/o}$ subunits in pulling force generation has also remained ambiguous (*Figure 1a*). Membrane-attached $G\alpha \cdot GDP$ associates with GoLoco motifs present in the homologous GPR-1/2, Pins, and LGN proteins (*Kimple et al., 2002*; *Schaefer et al., 2001*). This preference for the GDP-bound 'inactive' $G\alpha$ state explains why RGS-7, a putative GTPase activating protein (GAP), promotes spindle positioning (*Hess et al., 2004*). However, the role of another conserved regulator of $G\alpha$ signaling, RIC-8, remains poorly understood (*Afshar et al., 2004*; *Miller and Rand, 2000*; *Tall et al., 2003*). RIC-8 was shown to act as a guanine nucleotide exchange factor (GEF) in vitro, while it may function in vivo as a $G\alpha$ chaperone or as both a GEF and chaperone (*Afshar et al., 2004*; *Afshar et al., 2005*; *David et al., 2005*; *Gabay et al., 2011*; *Hampoelz et al., 2005*; *Tall et al., 2003*; *Wang et al., 2005*). In addition to RIC-8, G-protein coupled receptors and $G\alpha_o \cdot GTP$ signaling contribute to spindle positioning in *Drosophila* neuroblasts and sensory organ precursor cells (*Katanaev et al., 2005*; *Schaefer et al., 2001*; *Yoshiura et al., 2012*). Therefore, it has been proposed that the $G\alpha \cdot GTP$-binding and hydrolysis cycle forms a critical step in cortical pulling force generation (*Afshar et al., 2004*; *Srinivasan et al., 2003*; *Yoshiura et al., 2012*). However, it is difficult to distinguish whether $G\alpha_o \cdot GTP$ contributes to force generation, or more indirectly relays cell-cell signaling to the spindle.

Here, we describe an optogenetic strategy for the systematic examination of individual contributions of cortical pulling force components in vivo. We use the *C. elegans* one-cell embryo (P0), which undergoes reproducible spindle positioning and asymmetric cell division in the absence of cell-cell signaling (*Video 1*) (*Rose and Gonczy, 2014*). As an initial hurdle, modifying endogenous genes with tunable light-controlled interacting protein tags (TULIPs) induced strong germline silencing. We developed a strategy to promote expression of foreign sequences in the *C. elegans* germline, which

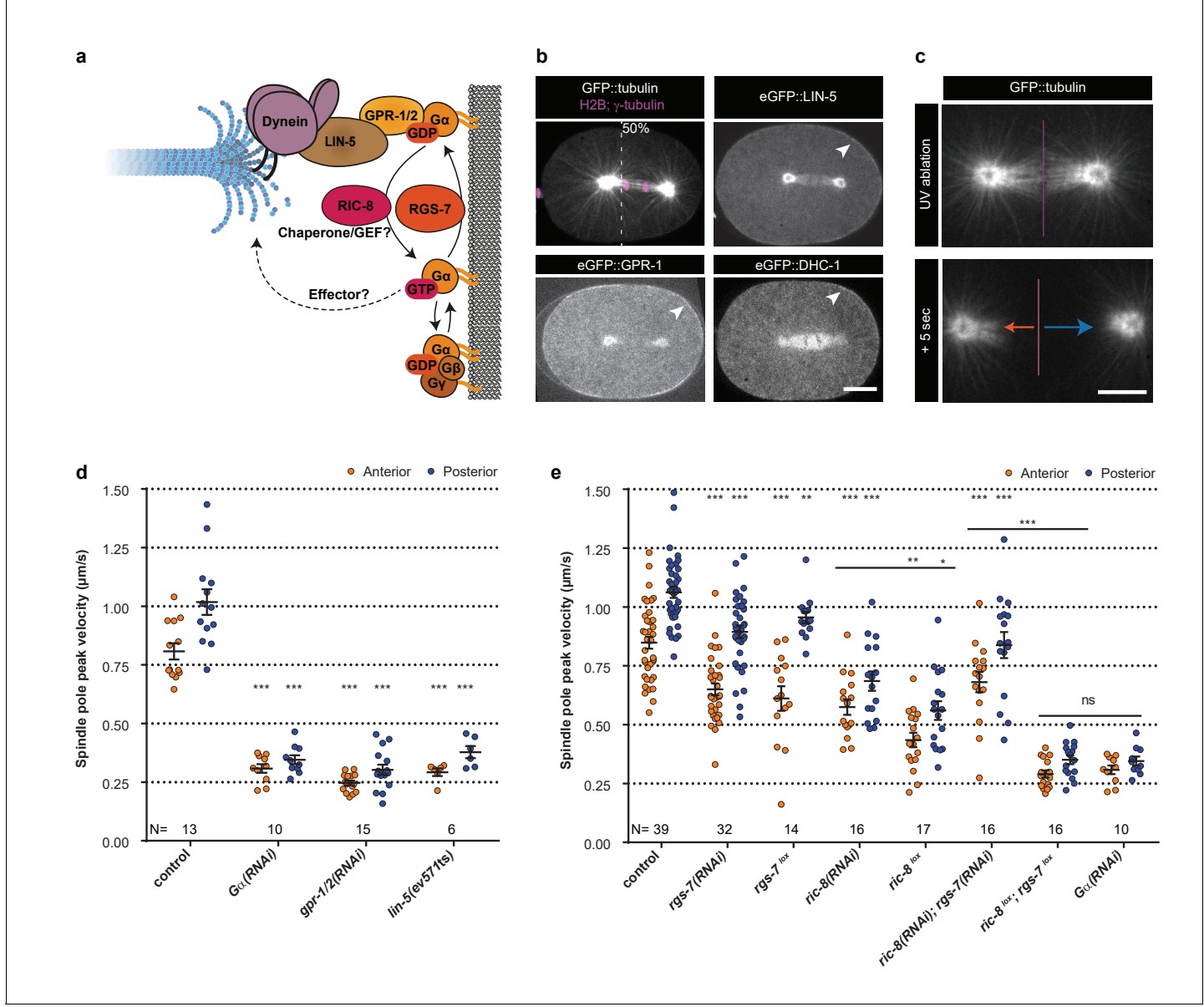

**Figure 1.** Gα regulation by RIC-8 and RGS-7 is essential for cortical pulling force generation. (a) Cartoon model representing mechanisms and functions discussed in the text. Gα•GDP–GPR-1/2–LIN-5–dynein anchors dynamic microtubule plus-ends and generates cortical pulling forces on the mitotic spindle. Gα•GDP can assemble a Gαβγ or Gα–GPR-1/2–LIN-5 trimer. The Gα•GDP/GTP nucleotide state is regulated by the GAP RGS-7. For RIC-8, functions as Gα GEF and chaperone are reported. Gα•GTP could promote spindle positioning through unknown downstream effectors. (b) Spinning disk confocal images of anaphase spindle positioning away from the cell center (dashed line) in the *C. elegans* zygote. The upper left panel shows the spindle with labeled tubulin and DNA. Other panels: endogenous GPR-1, LIN-5, and dynein (DHC-1) fused to eGFP are present in the cytoplasm, at the cell cortex (arrowheads), and spindle structures. Scale bar: 10 µm. (c) Spinning disk confocal images of the mitotic spindle (marked by GFP::tubulin). Upon UV-laser ablation of the spindle midzone (violet line), spindle poles separate with velocities that represent the respective net force acting on each pole (arrows). Scale bar: 5 µm. (d) Spindle pole peak velocities after midzone ablation. Control is the *gfp::tubulin* strain. Other conditions: inactivation of Gα, GPR-1/2, and LIN-5. Error bars: s.e.m. Welch's Student's t-test; ***p<0.001. (e) Spindle severing experiments in embryos where RIC-8 and RGS-7 were depleted by RNAi or induced tissue specific CRE-lox-mediated knockout of the endogenous gene (lox). Control is the *gfp::tubulin* strain, see *Figure 1—figure supplement 1* for knockout method and additional controls. Error bars: s.e.m. Welch's Student's t-test and Mann Whitney U test; *p<0.05, **p<0.01, ***p<0.001. See *Supplementary file 1* for detailed genotypes. Anterior is to the left in all microscopy images.

DOI: https://doi.org/10.7554/eLife.38198.003

The following figure supplements are available for figure 1:

**Figure supplement 1.** Method of inducible knockout of essential genes in the *C. elegans* germline.

DOI: https://doi.org/10.7554/eLife.38198.004

*Figure 1 continued on next page*

*Figure 1 continued*

**Figure supplement 2.** Inducible knockout of essential genes in the *C. elegans* germline.

DOI: https://doi.org/10.7554/eLife.38198.005

is based in part on a new codon usage adaptation method (GLO, GermLine Optimized). This allowed the light-controlled localization of endogenous proteins through ePDZ–LOV domain interactions in the early *C. elegans* embryo. Our results show that Gα•GDP and GPR-1/2 can be replaced with a light-inducible membrane anchor, while LIN-5 is required as activator of dynein-dependent cortical pulling force generation. Local light-controlled LIN-5 recruitment enabled us to manipulate the spindle position and orientation, and thereby the outcome of cell division in the early embryo.

## Results

### Germline-specific gene knockout reveals that RIC-8 and RGS-7 cooperate in positive Gα regulation and cortical pulling force generation

We set out to systematically investigate the individual roles of the proteins involved in cortical pulling force generation. Our previous studies and CRISPR/Cas9-assisted endogenous tagging demonstrated that cytoplasmic dynein and the Gα–GPR-1/2–LIN-5 complex overlap and function together in pulling force generation at the cell cortex of *C. elegans* early blastomeres (*Figure 1b*) (*Portegijs et al., 2016*; *Schmidt et al., 2017*; *van der Voet et al., 2009*). As a read-out for pulling forces, we measured spindle pole peak velocities after UV-laser ablation of the spindle midzone (*Grill et al., 2001*) (*Figure 1c* and *Video 2*). Interfering with Gα, GPR-1/2, or LIN-5 function abolished significant force generation, as previously reported (*Figure 1d*). RNA interference (RNAi) of *ric-8* or *rgs-7* by dsRNA injection resulted in partial loss of pulling forces (*Figure 1e*). Double *ric-8 (RNAi); rgs-7(RNAi)* did not further decrease pulling forces as might be expected when RIC-8 and RGS-7 both promote a critical GTPase cycle (*Hess et al., 2004*; *Srinivasan et al., 2003*). However, RNAi of *ric-8* and *rgs-7* is known to cause incomplete gene inactivation, which could also explain the limited defects. To circumvent this caveat, we set out to generate germline-inducible knock-out alleles, as *ric-8* and *rgs-7* null mutants produce no or very few viable progeny (*Hess et al., 2004*; *Reynolds et al., 2005*). To this end, we inserted *lox* sites in the endogenous *ric-8* and *rgs-7* loci by CRISPR/Cas9-assisted recombineering (*Figure 1—figure supplement 1*), and expressed the CRE recombinase specifically in the germline (*Figure 1—figure supplement 1*). Compared to the control without CRE activity, knockout embryos showed reduced spindle pole peak velocities (*ric-8^{lox}*: anterior −50% and posterior −48%; *rgs-7^{lox}*: anterior −29% and posterior −11%), similar to or more defective than the corresponding RNAi embryos (*Figure 1e* and *Figure 1—figure supplement 2*). Importantly, the double knock-out of *ric-8^{lox}*; *rgs-7^{lox}* showed much reduced spindle pole movements (anterior −68% and posterior −67%), thereby resembling *Gα(RNAi)* (*Figure 1e*). This indicates that RIC-8 and RGS-7 act independently, or partly redundant, as positive regulators of Gα.

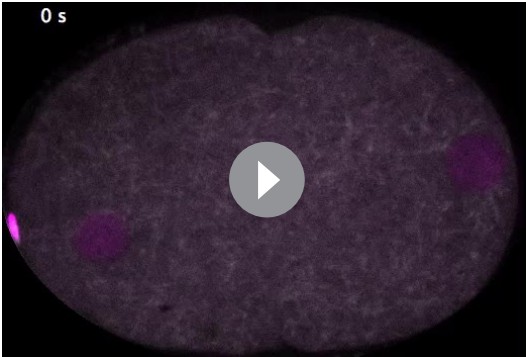

**Video 1.** Movie montage of mitosis in a one-cell *C. elegans* embryo expressing GFP::tubulin (greyscale, microtubules), mCherry::TBG-1 (magenta, centrosomes) and mCherry::HIS-48 (magenta, DNA). Images, which are single planes, were made as a time-lapse with one acquisition per 2 s and played back at 10 frames per second, with time point 0 being the final frame before the initiation of pronuclear meeting. Movie corresponds to the upper left panel in *Figure 1b*.

DOI: https://doi.org/10.7554/eLife.38198.006

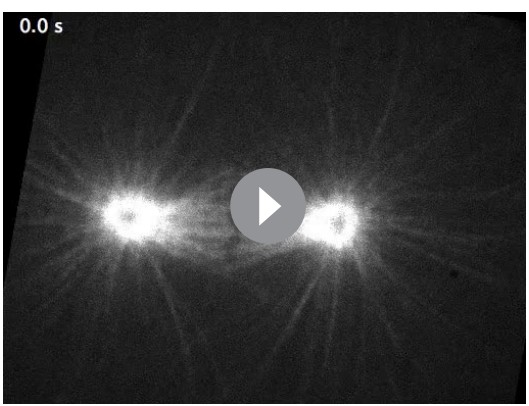

**Video 2.** Movie montage of a mitotic spindle severing assay in a one-cell *C. elegans* embryo expressing GFP:: tubulin (greyscale, microtubules). The spindle is severed at the onset of anaphase using a pulsed UV laser (not visible), after which centrosomes are separated with speeds proportional to the net forces acting on them. Images, which are single planes, were made as a streaming acquisition with 0.5 s of exposure and played back at 10 frames per second, with time point 0 between late metaphase and anaphase initiation. Movie corresponds to *Figure 1c*.
DOI: https://doi.org/10.7554/eLife.38198.007

## Germline-optimized codon adaptation promotes germline expression of transgenes

To gain further insight into the individual functions of cortical pulling force regulators, we sought to obtain spatiotemporal control of protein localization. To this end, we explored implementing the ePDZ–LOV system, which makes use of exposure to blue light to control protein heterodimerization (*Harterink et al., 2016*; *Strickland et al., 2012*). As introduction of *epdz*, *lov*, and *cre* sequences induced strong germline silencing responses, we developed a computational approach to design protein-coding sequences that are resistant to silencing in the germline. Our design algorithm assembles a coding sequence for any desired polypeptide from a list of 12-nucleotide words found in native germline-expressed genes (*Figure 2—figure supplement 1*). We hypothesized that transgenes designed in this way would mimic native genes and thereby evade the germline silencing machinery. Indeed, using this approach, we could obtain robust expression of several foreign transgenes that were otherwise silenced (*Figure 2—figure supplement 2*). Although most of these transgenes were stably expressed for many generations, two out of 16 distinct constructs tested showed evidence of gradual silencing when passaging strains in culture (*Figure 2—figure supplement 3*). Therefore, as a further buffer against silencing, we combined our germline-optimized exons with poly-AT-cluster rich intron sequences, which were recently demonstrated to protect against germline silencing (*Frøkjær-Jensen et al., 2016*; *Zhang et al., 2018*). This combined approach resulted in stable germline expression of eight out of eight transgenes and enabled implementation of the ePDZ–LOV system for use in the *C. elegans* early embryo.

## An optimed ePDZ-LOV system enables subcellular control of protein localization in the *C. elegans* early embryo

To characterize the ePDZ–LOV system, we created a strain with a membrane-bound LOV2 domain, expressed as a pleckstrin-homology domain (PH)–eGFP protein fusion (PH::LOV), together with cytosolic ePDZ::mCherry (*Figure 2a*). Illumination with a blue (491 nm) laser rapidly induced recruitment of ePDZ::mCherry to PH::LOV, and allowed both global and local cortical enrichment in one-cell embryos (*Figure 2b* and *Videos 3–5*). Because GFP is also excited with blue light, experiments that involve GFP imaging imply global and continuous induction of the ePDZ–LOV interaction. To test the reversibility of the ePDZ–LOV interaction, we followed ePDZ::mCherry membrane localization after a global activation pulse, and found dissociation kinetics similar to those reported by others (*Hallett et al., 2016*) (t½=42 s; *Figure 2c* and *Video 6*). Thus, we conclude that the ePDZ–LOV system is suitable for controlled protein localization in the early *C. elegans* embryo.

Next, we examined whether membrane recruitment of RGS-7 and RIC-8 promotes pulling forces. This could provide insight into the in vivo relevance of the Gα•GDP/GTP cycle and indicate whether RIC-8 is more likely to contribute as a chaperone or as a GEF. Regulation of the Gα•GDP/GTP cycle normally takes place at the cell membrane, while chaperoning of Gα folding and trafficking is expected to occur in the cytosol and at endomembranes (*Gabay et al., 2011*). We created strains expressing endogenous RIC-8 and RGS-7 as ePDZ::mCherry protein fusions. When combined with PH::LOV, this resulted in light-inducible membrane recruitment of RIC-8 and RGS-7 (*Figures 3a* and *4a*, and *Video 7*). Global cortical enrichment of RGS-7 enhanced spindle pole movements (anterior +25% and posterior +20%) (*Figure 3b*) and spindle oscillations (*Figure 3c*). The RGS-7::ePDZ

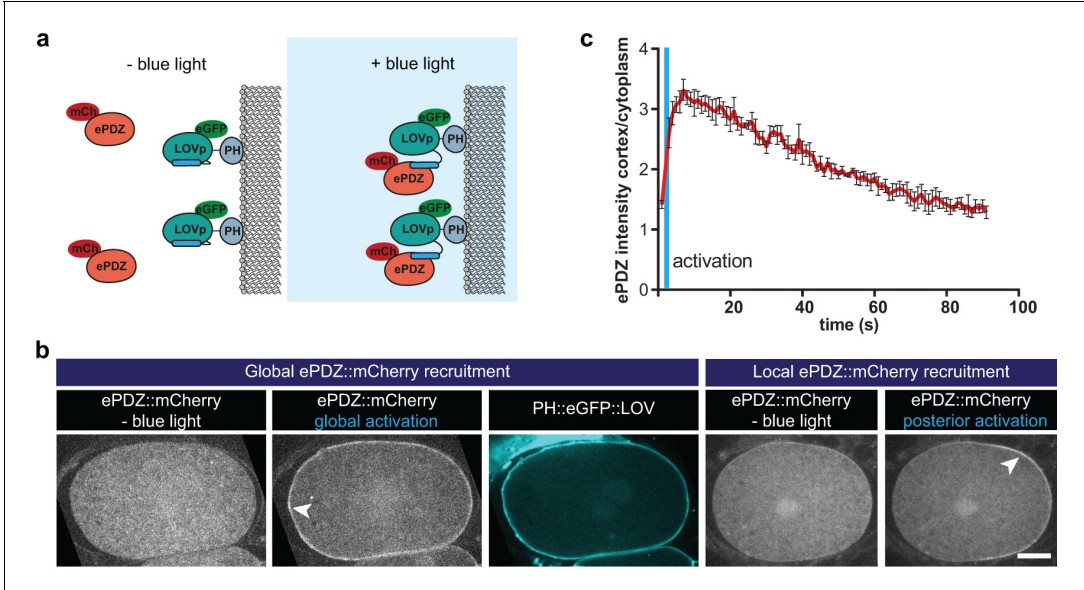

**Figure 2.** Optimized ePDZ–LOV enables light-inducible control of endogenous protein localization in the *C.elegans* one-cell embryo. (a) Cartoon model illustrating the proof of concept wherein cytosolic ePDZ::mCherry is cortically recruited to membrane PH::LOV upon activation with blue light. Blue fields indicate conditions in which both ePDZ- and LOV components are present, and an ePDZ–LOV interaction is induced with blue light. (b) Spinning disk confocal images showing light-controlled localization of proteins in the *C. elegans* zygote (arrowheads). See Materials and methods for the local activation procedure. Also see *Videos 3–5*. Scale bar: 10 µm. Anterior is to the left in all microscopy images. (c) Quantification of cortical ePDZ::mCherry enrichment measured over time after a 1 s pulse activation (blue vertical line). Error bars: s.e.m. $t^1/_2$ calculated with single component non-linear regression.

DOI: https://doi.org/10.7554/eLife.38198.008

The following figure supplements are available for figure 2:

**Figure supplement 1.** Method of germline-optimization of coding sequences confers resistance to silencing.
DOI: https://doi.org/10.7554/eLife.38198.009

**Figure supplement 2.** Germline-optimization of coding sequences confers resistance to silencing.
DOI: https://doi.org/10.7554/eLife.38198.010

**Figure supplement 3.** Germline-optimization can allow expression even of highly silencing-prone transgenes, although expression is transient.
DOI: https://doi.org/10.7554/eLife.38198.011

signal was too subtle to reliably control its local recruitment. As an alternative strategy, we fused eGFP::LOV to endogenous PAR-6, which localizes to the anterior cortex of the zygote (*Figure 3a*). Following global light exposure, recruitment of RGS-7::ePDZ to PAR-6::LOV enhanced the peak velocities of both spindle poles, but most significantly the movement of the anterior pole (anterior +25% and posterior +14%; *Figure 3b,c*). Thus, cortical recruitment of RGS-7 acutely increases pulling forces, in agreement with its proposed function as a GAP that promotes Gα•GDP–GPR-1/2 interaction. In contrast, cortical enrichment of RIC-8 did not significantly enhance pulling forces (*Figure 4b*). Thus, in agreement with the *ric-8^lox^; rgs-7^lox^* synergistic phenotype, our optogenetic localization experiments support a model in which RIC-8 and RGS-7 promote Gα function at different levels. While RGS-7 probably functions as a GAP, our data are in line with RIC-8 acting in vivo as a Gα chaperone, rather than a GEF, thus indirectly promoting force generation.

## Membrane anchoring of GPR-1^Pins/LGN^ in the absence of Gα reconstructs a cortical pulling force generator

To directly address whether Gα•GTP might contribute to spindle positioning and if Gα•GDP serves merely as a membrane anchor, we aimed to reconstruct a cortical force generator in the absence of Gα (*Figure 5a*). We obtained optogenetic control over the membrane localization of GPR-1 by combining endogenously labeled *epdz::mcherry::gpr-1* (ePDZ::GPR-1) with knockout of *gpr-2*, expression of PH::LOV and Gα RNAi (*Figure 5b,c* and *Videos 8* and *9*). Live imaging and immunohistochemistry confirmed light-induced cortical recruitment of ePDZ::GPR-1 and

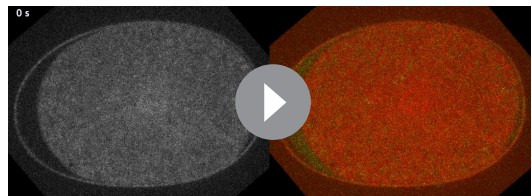

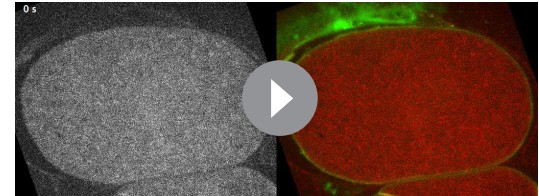

**Video 3.** Movie montage of a mitotic one-cell *C. elegans* embryo expressing diffuse cytosolic ePDZ:: mCherry (greyscale, left; red, right) without PH::eGFP:: LOV (green background, right). The movie shows diffuse localization of ePDZ::mCherry in the presence of global and continuous blue light exposure, but in absence of a cortical LOV anchor. Images, which are single planes, were made as a time-lapse with one acquisition per 2 s for both 568 nm and 491 nm illumination and played back at 10 frames per second, with time point 0 starting at metaphase. The acquisition in the 568 nm channel at time point 0 shows localization of ePDZ::mCherry in complete absence of blue light, as embryos were kept in the dark before image acquisition. Movie corresponds to no main figure, and serves as a control for *Video 4*.
DOI: https://doi.org/10.7554/eLife.38198.012

**Video 4.** Movie montage of a mitotic one-cell *C. elegans* embryo expressing diffuse cytosolic ePDZ:: mCherry (greyscale, left; red, right) and the membrane anchor PH::eGFP::LOV (green, right). The movie shows relocalization of diffuse ePDZ::mCherry to the cortex by global and continuousactivation of cortical LOV using blue light. Images, which are single planes, were made as a time-lapse with one acquisition per 2 s for both 568 nm and 491 nm illumination and played back at 10 frames per second, with time point 0 starting at late prophase. The acquisition in the 568 nm channel at time point 0 shows localization of ePDZ::mCherry in complete absence of blue light, as embryos were kept in the dark before image acquisition. Movie corresponds to the lower left panels in *Figure 2*.
DOI: https://doi.org/10.7554/eLife.38198.013

consequently LIN-5 (*Figure 5b* and *Figure 5—figure supplement 1*). Spindle movements appeared reduced following the tagging of *gpr-1* and knockout of *gpr-2* (anterior −40% and posterior −26%) (*Figure 5—figure supplement 2*). However, light-induced ePDZ::GPR-1 recruitment increased spindle pole movements (anterior +56% and posterior +10%) (*Figure 5d, f*). Moreover, membrane-localized ePDZ::GPR-1 sustained force generation in the absence of Gα (anterior +195% and posterior +232%), indicating that Gα is dispensable for cortical pulling force generation. Recruitment of ePDZ::GPR-1 restored spindle pole movements to a similar degree in *Gα(RNAi)* and *Gα(RNAi); ric-8(RNAi)* embryos (*Figure 5d*). Thus, cortical pulling forces can be generated when the Gα membrane anchor function is replaced by PH::LOV, and most likely in the absence of $G\alpha_{i/o} \bullet GTP$. We conclude that Gα functions as a membrane anchor and that $G\alpha \bullet GTP$ does not perform an essential function in pulling force generation.

Light-controlled heterodimerization exhibits a certain level of dark state activity (*Hallett et al., 2016*). We performed spindle severing experiments in the absence of blue light to confirm the light-specificity of recruitment. Replacement of *gfp::tubulin* with an *mcherry::tubulin* transgene allowed for tracking of the spindle in the absence of blue light and consequently LOV activation. We observed that the scattering of UV-light (355 nm) during midzone ablation also uncages the LOV domain. Nevertheless, the presence of blue light resulted in substantially elevated spindle pulling forces when compared to dark state experiments (anterior +33% and posterior +35%) (*Figure 5e*). We conclude that the observed spindle pole movements are light-dependent and the specific result of inducible cortical recruitment of GPR-1.

Considering that spindle poles moved in three dimensions after recruitment of GPR-1, measuring peak velocities after midzone severing by tracking the poles in two dimensions likely underestimates the resulting pulling forces. Therefore, we used an additional read-out of cortical pulling forces. Cortical pulling events cause inward movements of the plasma membrane, which are visible by spinning disk-confocal fluorescence microscopy as extended invaginations of the plasma membrane (*Redemann et al., 2010*) (*Figure 5g* and *Videos 10* and *11*). These membrane invaginations occur in wild type embryos, depend on microtubules and cortical force generator components, and correlate with the distribution of pulling force generators (*Redemann et al., 2010*). Therefore, these membrane invaginations most likely reveal the presence and distribution of active individual force generators.

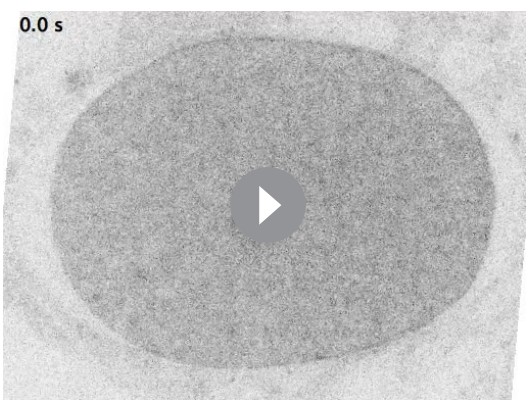 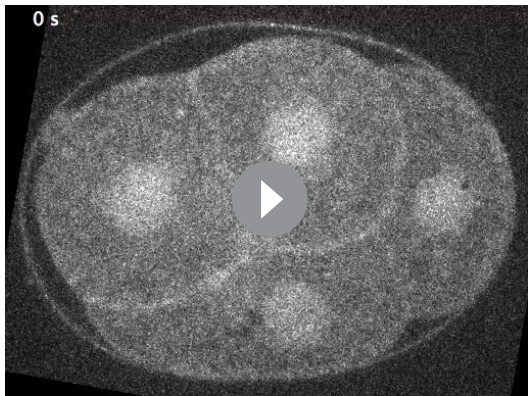

**Video 5.** Movie montage of a mitotic one-cell *C. elegans* embryo expressing diffuse cytosolic ePDZ::mCherry (inverted greyscale) and the membrane anchor PH::eGFP::LOV (not shown). The movie shows relocalization of diffuse ePDZ::mCherry to the posterior cortex by local activation of cortical LOV using low-intensity blue light. Activation of the ePDZ–LOV2 interaction is induced at the posterior cortex using local illumination with a 491 nm laser. The embryo was otherwise shielded from blue light before and during the experiment. Images, which are single planes, were made as a streaming acquisition with 0.5 s of exposure and played back at 10 frames per second, with time point 0 corresponding to late prophase. Movie corresponds to the lower right panels in *Figure 2*.
DOI: https://doi.org/10.7554/eLife.38198.014

**Video 6.** Movie montage of a four-cell *C. elegans* embryo expressing diffuse cytosolic ePDZ::mCherry (greyscale) and the membrane anchor PH::eGFP::LOV (not shown). The movie shows relocalization of diffuse ePDZ::mCherry to the cortex by activation of cortical LOV using a single pulse of blue light, and subsequent return to the dark state in the absence of blue light. Images, which are single planes, were made as a time-lapse with one acquisition per 2 s played back at 10 frames per second, where time point 0 is the last acquisition before a single 1 s global pulse of 491 nm light. The acquisition at time point 0 shows localization of ePDZ::mCherry in complete absence of blue light, as the embryo was kept in the dark before and after global induction of the LOV–ePDZ interaction. Movie corresponds to *Figure 2c*.
DOI: https://doi.org/10.7554/eLife.38198.015

Using spinning disk confocal microscopy, we quantified membrane invaginations by counting the number of transient cortical PH::GFP dots in the sub-cortical plane. Control PH::GFP embryos showed on average 138 membrane invaginations during anaphase in an area covering approximately $\frac{1}{3}$ of the cell surface (*Figure 5g*). When plotted along the anterior-posterior axis, the distribution of these invaginations reflected the three described cortical domains: anterior, posterior, and a posterior lateral region at ±60% embryo length (*Rose and Gonczy, 2014*) (*Figure 5h*-left). The posterior lateral band region localizes the LET-99 DEP-domain protein, which antagonizes the localization of GPR-1/2 and thereby pulling force generation (*Krueger et al., 2010*; *Tsou et al., 2003*). This explains the absence of invaginations around 60% embryo length (*Figure 5h*-left).

Cortical GPR-1 recruitment resulted in a total number of 174 (+25% compared to *ph::lov* control) invaginations in the presence, and 122 invaginations (+249% compared to *Gα(RNAi)* embryos) in the absence of Gα (*Figure 5* and *Figure 5—figure supplement 3*). Thus, in agreement with our observations in spindle severing assays, GPR-1 recruitment to the membrane induces cortical pulling forces, even in the apparent absence of Gα proteins. The lack of invaginations around 60% embryo length was no longer detected when ePDZ::GPR-1 was recruited to PH::LOV. In agreement, the characteristic dip in cortical GPR-1 localization (e.g.: *Figure 1b*) was no longer detected after ectopic GPR-1 recruitment (*Figure 5b*). Thus, as expected, LET-99 does not antagonize the cortical recruitment of ePDZ::GPR-1 by PH::LOV, in contrast to the Gα•GDP-mediated localization of endogenous GPR-1/2. The pattern of invaginations still showed two peaks and a mild dip at 50% embryo length (*Figure 5h*). The remaining peak numbers of invaginations likely represent the cortical regions closest to the spindle poles, as these sites contact the highest numbers of astral microtubules. Taken together, Gα is not essential for force generation, but the characteristic distribution of force generating events is likely regulated in part at the Gα protein or Gα–GPR-1/2 protein interaction level.

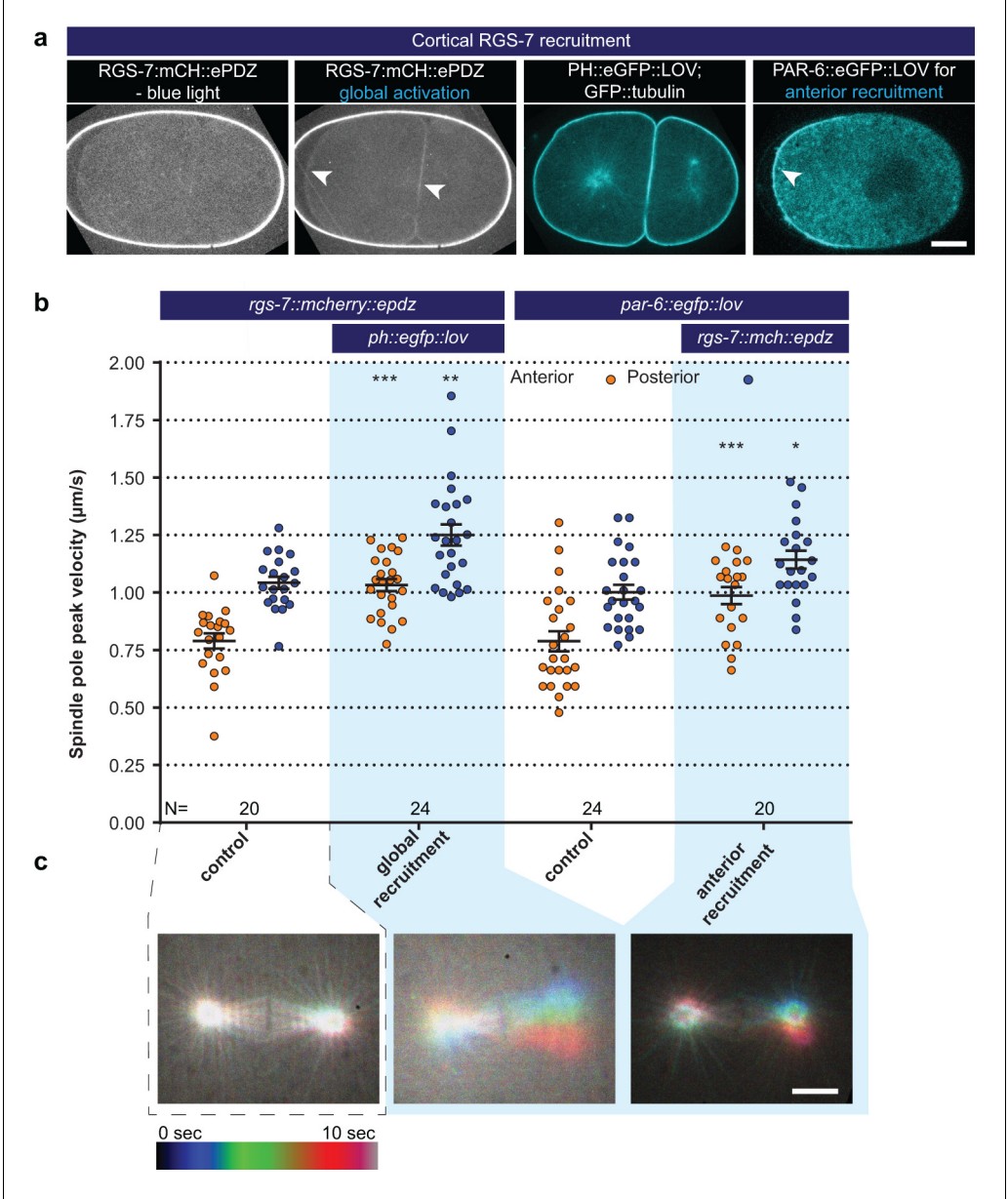

**Figure 3.** Light-controlled localization of endogenous Gα regulator RGS-7 in the *C. elegans* embryo. (a) Light-controlled localization of endogenous RGS-7 to membrane PH::LOV (arrowheads, note that the eggshell shows strong autofluorescence in the red channel). Most right panel: anterior localization of PAR-6::eGFP::LOV. Scale bar: 10 µm (b) Spindle severing experiments after light-induced cortical localization of RGS-7 (blue fields). Controls are the *rgs-7::mcherry::epdz* and *par-6::egfp::lov* strains. Experimental conditions: combination with *ph::egfp::lov* and *rgs-7::mcherry::epdz*. Blue fields indicate conditions in which both ePDZ and LOV components are present, and an ePDZ–LOV interaction is induced with blue light. Blue light activation was global and continuous. Error bars: s.e.m. Welch's Student's t-test; *p<0.05, **p<0.01, ***p<0.001. (c) Maximum projections of spindle movements for 10 s using a temporal color coding scheme to visualize spindle movements. A stationary spindle produces a white maximum projection, whereas a mobile spindle leaves a colored trace. Scale bar: 5 µm.
DOI: https://doi.org/10.7554/eLife.38198.016

## Direct cortical anchoring of dynein is insufficient for cortical pulling force generation

The Gα–GPR-1/2–LIN-5 complex has been suggested to function as a dynein anchor (*Kotak et al., 2012*; *di Pietro et al., 2016*). Our optogenetic approach allows replacing the entire complex by

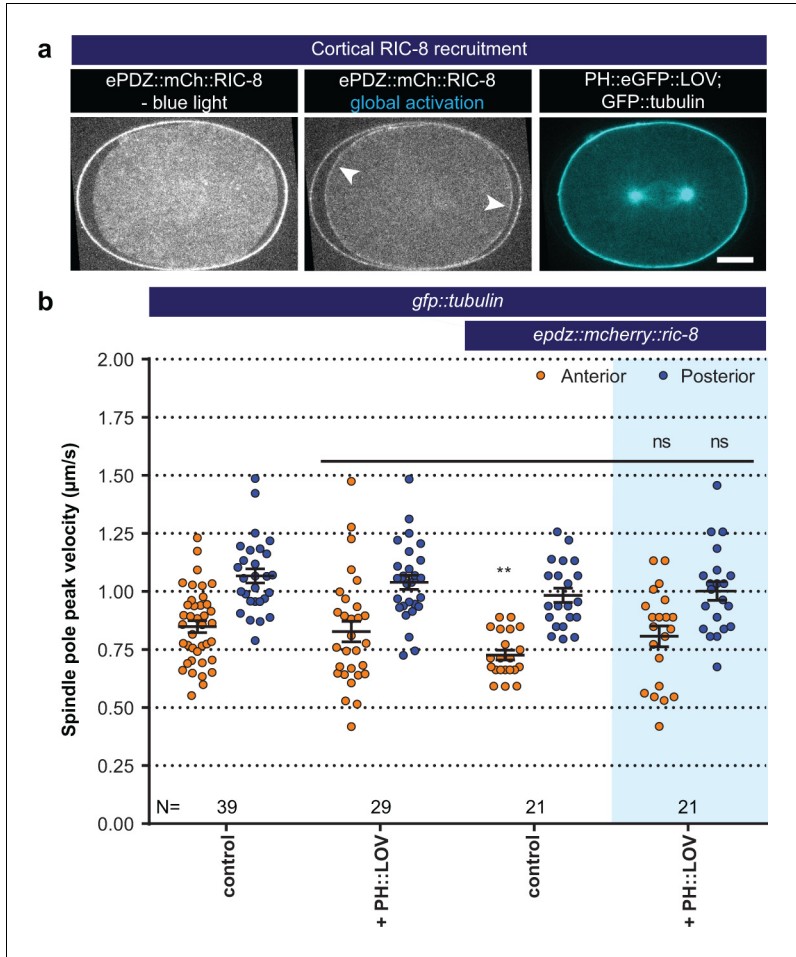

**Figure 4.** Light-controlled localization of the endogenous Gα regulator RIC-8 in the *C. elegans* embryo. (**a**) Light-controlled localization of endogenous RIC-8 to membrane PH::eGFP::LOV (arrowheads, note that the eggshell shows strong autofluorescence in the red channel). Scale bar: 10 μm. (**b**) Spindle severing experiments after light-induced cortical localization of RIC-8 (blue fields). Controls are *gfp::tubulin* and *epdz::mcherry::ric-8* strains. Experimental conditions: combination with *ph::egfp::lov*. Blue fields indicate conditions in which both ePDZ and LOV components are present, and an ePDZ–LOV interaction is induced with blue light. Error bars: s.e.m. Welch's Student's t-test; ns p>0.05. See **Supplementary file 1** for detailed genotypes. Anterior is to the left in all microscopy images.

DOI: https://doi.org/10.7554/eLife.38198.017

PH::LOV, and examining whether the complex strictly acts as an anchor, or whether individual components have additional functions (**Figure 6a**). To directly recruit dynein to the cortex, we generated an *epdz::mcherry* knock-in allele of *dhc-1* (dynein heavy chain). While homozygous *epdz::mcherry:: dhc-1* (ePDZ::DHC-1) was viable, its combination with *ph::egfp::lov* was lethal. This effect was also observed for an ePDZ::GFP fusion of DHC-1 in the presence of PH::LOV, but not in the absence of PH::LOV or for mCherry::DHC-1 without the ePDZ domain. Therefore, we attributed the lethality to ePDZ–LOV dark state interactions that disturb essential dynein functions. We circumvented this effect by using *epdz::mcherry::dhc-1* in combination with a wild-type allele (*epdz::mcherry::dhc-1/+*) to control dynein localization links to different figure (**Figure 6b,c** and **Videos 12** and **13**). We found that induced ePDZ::DHC-1 cortical recruitment in the presence of the wild type complex slightly (but not significantly) increased spindle pole movements (anterior +10% and posterior +5%, 162 membrane invaginations:+17%; **Figure 6d–f**). In addition, cortical recruitment of ePDZ::DHC-1 slightly increased the nearly absent spindle pole movements in *lin-5(RNAi)* embryos (**Figure 6—figure supplement 2**). Notably, however, cortical ePDZ::DHC-1 recruitment in the absence of a wild type

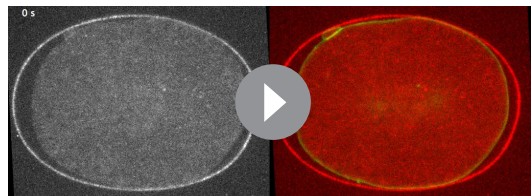

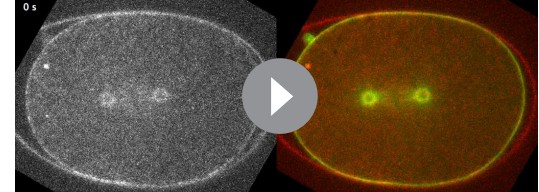

**Video 7.** Movie montage of a mitotic one-cell *C. elegans* embryo expressing endogenously labeled ePDZ::mCherry::RIC-8 (greyscale, left; red, right), the membrane anchor PH::eGFP::LOV and GFP::tubulin (both green, right). The movie shows relocalization ePDZ::mCherry::RIC-8 to the cortex by global and continuous activation of cortical LOV using blue light. Images, which are single planes, were made as a time-lapse with one acquisition per 2 s for both 568 nm and 491 nm illumination and played back at 10 frames per second, with time point 0 starting at late prophase. The acquisition in the 568 nm channel at time point 0 shows localization of ePDZ::mCherry::RIC-8 in complete absence of blue light, as embryos were kept in the dark before image acquisition. Movie corresponds to citation links to different figure *Figure 3c*.
DOI: https://doi.org/10.7554/eLife.38198.018

**Video 8.** Movie montage of a mitotic one-cell *C. elegans* embryo expressing endogenously labeled ePDZ::mCherry::GPR-1 (greyscale, left; red, right) in a Δ*gpr-2* genetic background, the membrane anchor PH::eGFP::LOV and GFP::tubulin (both green, right). The movie shows relocalization of ePDZ::mCherry::GPR-1 to the cortex by global and continuous activation of cortical LOV using blue light. Images, which are single planes, were made as a time-lapse with one acquisition per 2 s for both 568 nm and 491 nm illumination and played back at 10 frames per second, with time point 0 corresponding with early metaphase. The acquisition in the 568 nm channel at time point 0 shows localization of ePDZ::mCherry::GPR-1 in complete absence of blue light, as embryos were kept in the dark before image acquisition. Movie serves as a control to *Video 9*.
DOI: https://doi.org/10.7554/eLife.38198.023

complex (G*α*(RNAi), gpr-1/2(RNAi) or *lin-5(RNAi)* embryos) did not result in substantial pulling force generation, spindle movements, or membrane invaginations (*Figure 6d–f* and *Figure 6—figure supplement 1*). Because direct cortical dynein anchoring does not support force generation, it is likely that the LIN-5 complex performs essential functions beyond providing a structural dynein anchor.

## Cortical LIN-5[Mud/NuMA] is essential and sufficient for dynein-dependent pulling force generation

In vitro reconstitution studies established that homodimerizing adapters containing extended coiled-coil domains are critical for dynein activity (*McKenney et al., 2014*; *Schlager et al., 2014*). LIN-5 and its homologs NuMA and Mud are predicted to contain a long coiled-coil domain, to homodimerize, and to interact with dynein (*Kotak et al., 2012*; *Lorson et al., 2000*; *Merdes et al., 1996*). To investigate if LIN-5 can activate dynein-dependent force generation, we recruited endogenous LIN-5 to the cortex (*Figure 7a–c* and *Videos 14* and *15*). Spindle severing experiments and invagination counting revealed that cortical LIN-5 recruitment greatly increased spindle pulling forces in otherwise wild type embryos (anterior +131% and posterior +68%, 557 invaginations:+303%) (*Figure 7d,e* and *Figure 7—figure supplement 1*). *gpr-1/2(RNAi)* embryos also showed strong dynein-dependent forces after cortical recruitment of LIN-5 (anterior +183% and posterior +244%, 429 invaginations:+1488%). In fact, cortical LIN-5 recruitment generated extreme premature pulling forces (*Video 16*) that could result in separation of centrosomes and their associated pronuclei even before formation of a bipolar spindle (*Video 17*). Therefore, embryos were kept in the absence of blue light until mitotic metaphase. Subsequent blue light exposure induced cortical LIN-5 recruitment within seconds, and the spindles showed excessive movements in all three dimensions well before cortical LIN-5 reached peak levels (*Figure 7c,e*, *Videos 14* and *15*). Therefore, the number of membrane invaginations in anaphase probably reflects the pulling forces more accurately than the average peak velocities of the poles (*Figure 7f* and *Figure 7—figure supplement 1*). These results identify LIN-5 as a strong activator of dynein in the generation of cortical pulling forces.

We observed similarly high pulling forces upon cortical LIN-5 recruitment in *gpr-1/2(RNAi)* and G*α(RNAi)* embryos (*Figure 7—figure supplement 1*). Thus, neither G*α* nor GPR-1/2 are required for force generation, and this particular context did not reveal a positive contribution of G*α•*GTP either. In immunofluorescence staining experiments, we observed that cortical recruitment of LIN-5 localized dynein to the cortex (*Figure 7—figure supplement 2*). Notably, the reverse was also seen: we

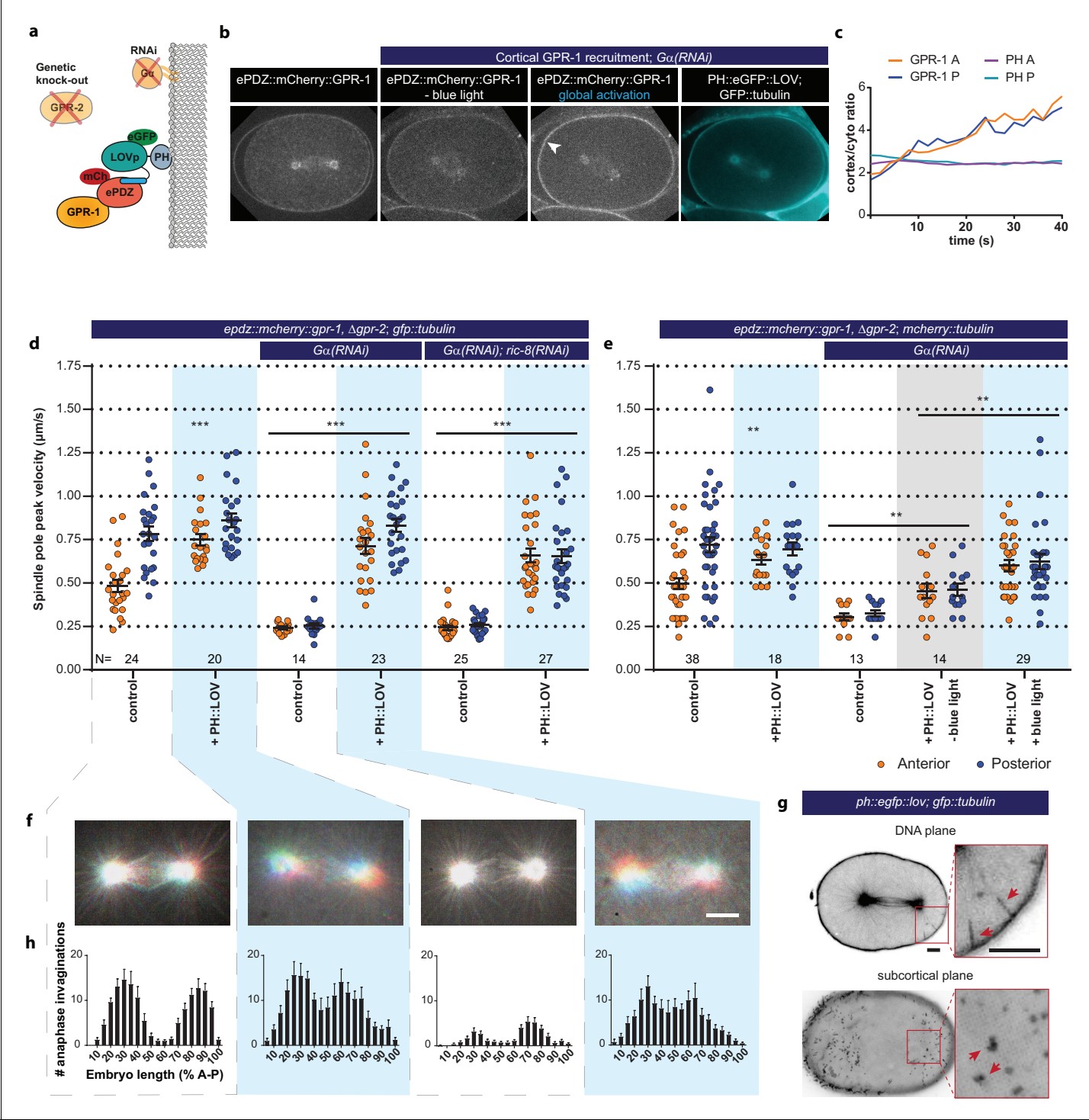

**Figure 5.** Light-inducible GPR-1 recruitment to the cortex rescues pulling force generation in the absence of Gα. (**a**) Cartoon model illustrating the experiment that localizes GPR-1 directly to the membrane, bypassing the wild type membrane anchor Gα which is inactivated by RNAi. (**b**) Spinning disk confocal images of light-controlled cortical GPR-1 recruitment independent of the wild type anchor Gα (arrowheads; note the autofluorescent eggshell in the mCherry channel). Scale bar: 10 μm. (**c**) Quantification of cortical GPR-1 recruitment during continuous activation of the ePDZ–LOV interaction, represented as the ratio of cortical/cytoplasmic signal. Also see *Videos 8* and *9*. (**d**) Spindle severing experiments in combination with cortical recruitment of endogenous GPR-1. Control is the *epdz::mcherry::gpr-1, Δgpr-2; gfp::tubulin* strain. Experimental conditions: combinations with *ph::egfp::lov*, *Gα(RNAi)*, and *Gα(RNAi); ric-8(RNAi)*. Blue fields indicate conditions in which both ePDZ- and LOV components are present, and an ePDZ–LOV interaction is induced with blue light. Blue light activation was global and continuous. Error bars: s.e.m. Welch's Student's t-test and Mann Whitney U test; **p<0.01, ***p<0.001. (**e**) Spindle severing experiments in combination with cortical recruitment of endogenous GPR-1. Control is the

*Figure 5 continued on next page*

*Figure 5 continued*

*epdz::mcherry::gpr-1, Δgpr-2; mcherry::tubulin* strain. Experimental conditions: combinations with *ph::egfp::lov, Gα(RNAi)*, and the absence of blue light (grey field). Blue fields indicate conditions in which an ePDZ- and LOV component are present, and an ePDZ–LOV interaction is induced using blue light. Blue light activation was global and continuous. Error bars: s.e.m. Welch's Student's t-test and Mann Whitney U test; **p<0.01, ***p<0.001. (f) Maximum projections of spindle movements for 10 s using a temporal color coding scheme to visualize spindle movement as in *Figure 3c*. (g) Plasma membrane invaginations resulting from cortical pulling forces are visible as lines in the DNA plane and dots in the subcortical plane (red arrows). Larger structures are membrane ruffles, which are distinct from the more dynamic invaginations, as can be seen in *Video 11*. Scale bar: 5 μm. (h) Distribution of anaphase membrane invaginations plotted along anterior-posterior embryo length. Conditions were the same as for the connected experiments in d and f, except for the control, which was the *ph::egfp::lov; gfp::tubulin* strain and not *epdz::mcherry::gpr-1, Δgpr-2*. Scale bar: 5 μm. Blue fields indicate conditions in which ePDZ and LOV components are present, and an ePDZ–LOV interaction is induced using blue light. See *Supplementary file 1* for detailed genotypes. Anterior is to the left in all microscopy images.

DOI: https://doi.org/10.7554/eLife.38198.019

The following figure supplements are available for figure 5:

**Figure supplement 1.** Cortical GPR-1 recruitment subsequently recruits LIN-5 to the cortex.
DOI: https://doi.org/10.7554/eLife.38198.020

**Figure supplement 2.** Spindle severing experiments of CRISPR/Cas9-mediated *gpr-1* and *gpr-2* knockout alleles, and tagged *gpr-1*.
DOI: https://doi.org/10.7554/eLife.38198.021

**Figure supplement 3.** Cortical GPR-1 recruitment increases cortical pulling forces and the number of plasma membrane invaginations.
DOI: https://doi.org/10.7554/eLife.38198.022

detected LIN-5 at the cortex following the direct recruitment of ePDZ::DHC-1 to PH::LOV, even after knockdown of *gpr-1/2* by RNAi (*Figure 6—figure supplement 2*, top). The p150(Glued) dynactin subunit DNC-1 was also present at the cell cortex of such embryos, which indicates that at least some PH::LOV-localized dynein complexes contain the dynactin cofactor (*Figure 6—figure supplement 2*, lower panels). DNC-1[p150] colocalized with PH::LOV membrane-recruited dynein even in *lin-5(RNAi)* embryos. Thus, the cortical localization of dynein through a direct PH::LOV interaction leads to co-recruitment of LIN-5 and dynactin, but not to significant force generation. It is possible that these complexes adapt an inactive conformation or lack specific subunits of the dynein-dynactin motor complex. In contrast, dynein anchored at the cell cortex through the LIN-5 intermediate generated strong pulling forces. Together, these data indicate that association with membrane-attached LIN-5 is essential for dynein to generate cortical pulling forces.

## Local LIN-5[Mud/NuMA] recruitment provides experimental control over spindle and cell cleavage plane positioning

Next, we examined whether we could deploy cortical LIN-5 to manipulate spindle positioning and the outcome of cell division by local illumination with blue light. In the normal P0 cell, the spindle is displaced towards the posterior and cell cleavage creates a larger anterior blastomere (AB) and a smaller posterior blastomere (P1). Local recruitment of LIN-5 to the anterior cortex from metaphase onwards caused the P0 spindle to position anteriorly, which inverted the AB:P1 size asymmetry after cell division (*Figure 8a* and *Video 18*). In addition, recruiting LIN-5 laterally induced a completely perpendicular spindle position (*Figure 8b* and *Video 19*). While this triggered some furrowing at the anterior and posterior cell poles, the spindle switched back to an anterior-posterior orientation during

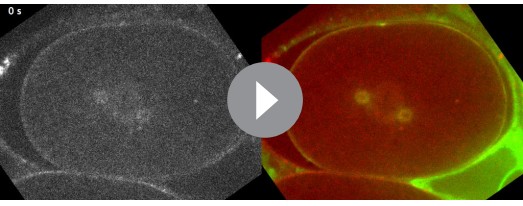

**Video 9.** Movie montage of a mitotic one-cell *C. elegans* embryo treated with Gα RNAi expressing endogenously labeled ePDZ::mCherry::GPR-1 (greyscale, left; red, right) in a Δgpr-2 genetic background, the membrane anchor PH::eGFP::LOV and GFP::tubulin (both green, right). The movie shows relocalization of ePDZ::mCherry::GPR-1 to the cortex by global and continuousactivation of cortical LOV using blue light. Images, which are single planes, were made as a time-lapse with one acquisition per 2 s for both 568 nm and 491 nm illumination and played back at 10 frames per second, with time point 0 corresponding with early metaphase. The acquisition in the 568 nm channel at time point 0 shows localization of ePDZ::mCherry::GPR-1 in complete absence of blue light, as embryos were kept in the dark before image acquisition. Movie corresponds to *Figure 5b*.
DOI: https://doi.org/10.7554/eLife.38198.024

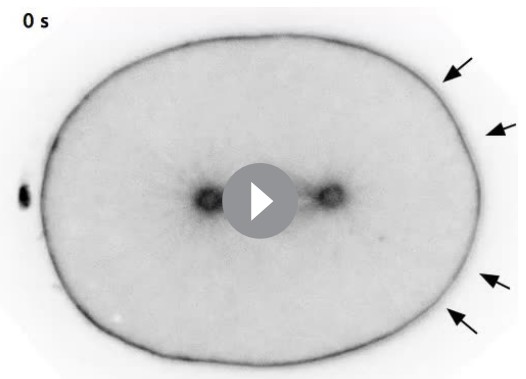

**Video 10.** Movie montage of a mitotic one-cell *C. elegans* embryo expressing GFP::tubulin (greyscale, microtubules) and PH::eGFP::LOV (greyscale, membrane). Invaginations (black arrows) are visible at the embryo membrane most pronouncedly in the posterior during late metaphase and anaphase. Images, which are single planes, were made as a streaming acquisition with 0.5 s of exposure and played back at 10 frames per second, with time point 0 between late metaphase and anaphase initiation. Movie corresponds to *Figure 5g*.
DOI: https://doi.org/10.7554/eLife.38198.025

**Video 11.** Movie montage of the subcortical area of a mitotic one-cell *C. elegans* embryo expressing GFP:: tubulin (inverted greyscale, microtubules), PH::eGFP:: LOV (inverted greyscale, membrane). Invaginations are visible as dots protruding inwards from the embryo membrane during late metaphase and anaphase. Images, which are single planes, were made as a streaming acquisition with 0.25 s of exposure and played back at 10 frames per second, with time point 0 corresponding with anaphase, 50 s before telophase initiation. Movie corresponds to *Figure 5g*.
DOI: https://doi.org/10.7554/eLife.38198.026

cytokinesis, possibly resulting from geometric constraints. We therefore switched to two-cell embryos with the relatively round AB and P1 blastomeres. In two-cell *gpr-1/2(RNAi)* embryos, the spindle fails to rotate in P1, resulting in a transverse spindle orientation in both blastomeres (*Srinivasan et al., 2003*). Importantly, local recruitment of LIN-5 to the membranes between AB and P1 promoted anterior-posterior spindle orientations in both blastomeres of *gpr-1/2(RNAi)* embryos (*Figure 8c* and *Videos 20* and *21*). These spindles maintained their anterior-posterior orientation throughout mitosis and induced cleavage furrows that reproducibly followed the spindle position. These experiments underline the determining role of LIN-5-dependent cortical pulling in spindle orientation and cell cleavage plane determination.

## Discussion

Recent advances in CRISPR/Cas9-mediated genome engineering and optogenetics hold far-reaching potential for cell and developmental biology (*Johnson and Toettcher, 2018*; *Waaijers and Boxem, 2014*). We combined these strategies to systematically control the localization of endogenous proteins in the *C. elegans* early embryo by light-induced ePDZ–LOV heterodimerization, to determine their individual contributions in spindle positioning. This demonstrated that LIN-5, but not Gα, RIC-8 and GPR-1/2, is intrinsically required for dynein-dependent pulling force generation. Based on our quantitative analyses, we propose that Gα–GPR-1/2 provides a regulatable membrane anchor, while membrane-bound LIN-5 acts as an obligate adapter and activator of cytoplasmic dynein at the cell cortex.

Our observations fit with and expand on those from studies in which the LIN-5-related Mud and NuMA proteins where tethered to the cortex. In two earlier studies, this resulted in the conclusion that the NuMA complex acts as a dynein anchor, but substitution of components or Gα protein removal was not tested (*Kotak et al., 2012*; *Ségalen et al., 2010*). However, a very recent study using human carcinoma cells followed an optogenetic strategy similar to ours, and also observed that a membrane anchor cannot replace the entire Gα–LGN–NuMA complex (*Okumura et al., 2018*). As in *C. elegans* embryos, a CAAX membrane anchor could substitute for Gα–LGN in human carcinoma cells, while dynein needed to be anchored through NuMA in order to generate spindle positioning forces (*Okumura et al., 2018*). Thus, observations in two different systems

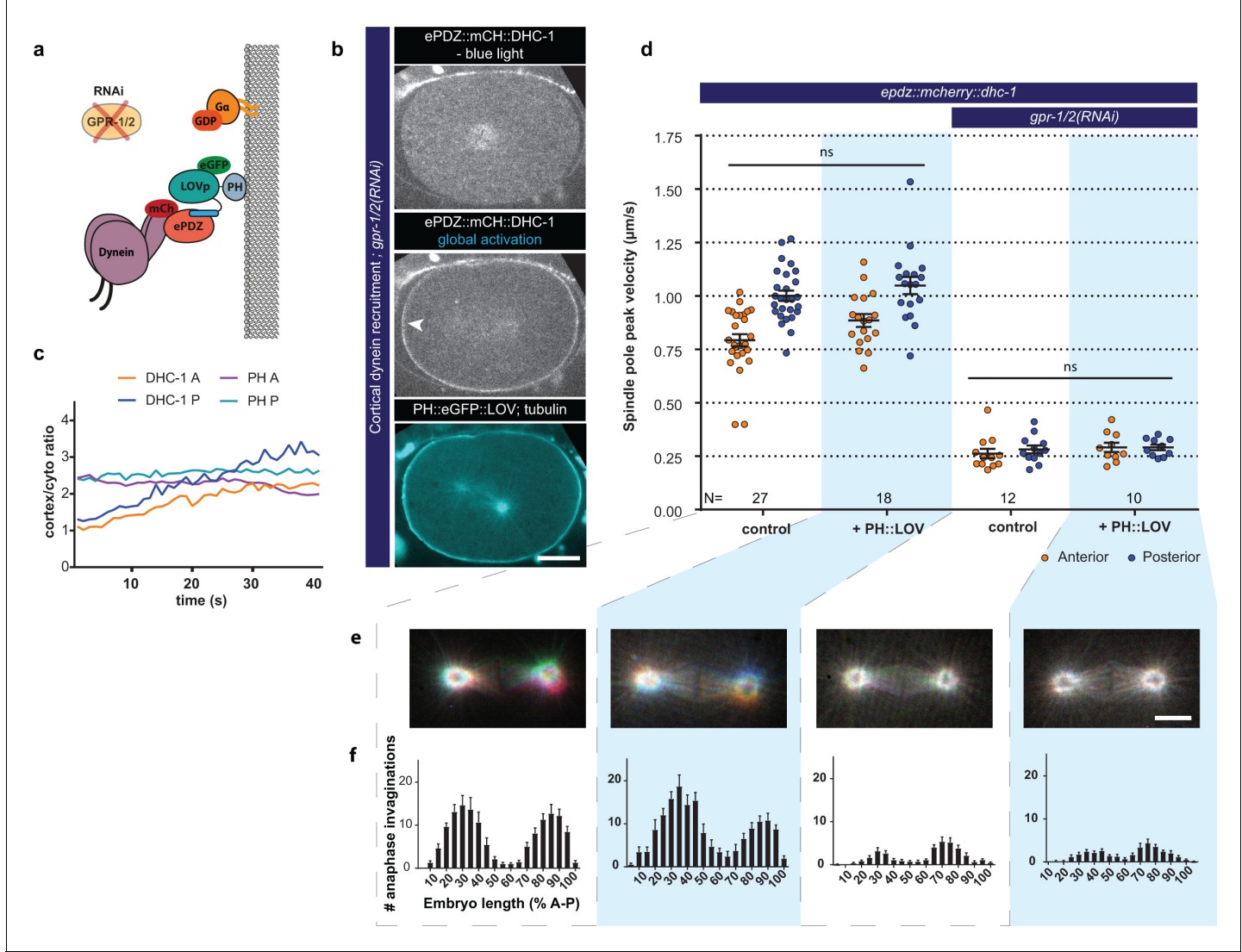

**Figure 6.** Direct cortical anchoring of dynein (DHC-1) is insufficient for cortical pulling force generation. (**a**) Cartoon model illustrating the experiment where dynein is recruited directly to the cortex. The wild type force generator complex is inactivated by RNAi. (**b**) Spinning disk confocal images showing light-controlled recruitment of dynein to the cortex (arrowheads, note the autofluorescent eggshell in the mCherry channel). Scale bar: 10 μm. (**c**) Quantification of cortical dynein recruitment during continuous activation of PH::LOV with blue light. (**d**) Spindle severing experiments with cortical dynein recruitment. Control is the *epdz::mcherry::dhc-1; gfp::tubulin* strain. Experimental conditions: combination of *ph::egfp::lov* and *gpr-1/2(RNAi)*. Blue light activation was global and continuous. Error bars: s.e.m. Welch's Student's t-test and Mann Whitney U test; ns p>0.05. (**e**) Maximum projections of spindle movements for 10 s using a temporal color coding scheme to visualize spindle movement as in *Figure 3c*. Scale bar: 5 μm. (**f**) Distribution of anaphase membrane invaginations plotted along anterior-posterior embryo length. Conditions were the same as for the connected experiments in d and e, except for the control, which was the *ph::egfp::lov; gfp::tubulin* strain and not *epdz::mcherry::dhc-1*. Blue fields indicate conditions in which both ePDZ and LOV components are present, and an ePDZ–LOV interaction is induced with blue light. See *Supplementary file 1* for detailed genotypes. Anterior is to the left in all microscopy images.

DOI: https://doi.org/10.7554/eLife.38198.027

The following figure supplements are available for figure 6:

**Figure supplement 1.** Cortical dynein (DHC-1) recruitment does not enhance cortical pulling force generation.
DOI: https://doi.org/10.7554/eLife.38198.028

**Figure supplement 2.** Cortical dynein (DHC-1) recruitment localizes LIN-5.
DOI: https://doi.org/10.7554/eLife.38198.029

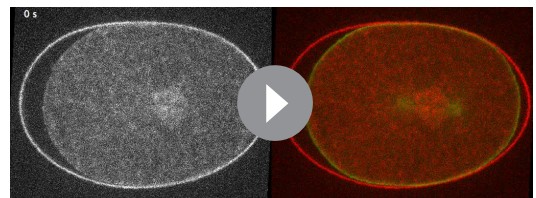

**Video 12.** Movie montage of a mitotic one-cell *C. elegans* embryo expressing endogenously labeled ePDZ::mCherry::DHC-1 (greyscale, left; red, right), the membrane anchor PH::eGFP::LOV and GFP::tubulin (both green, right). The movie shows relocalization of ePDZ::mCherry::DHC-1 to the cortex by global and continuousactivation of cortical LOV using blue light. Images, which are single planes, were made as a time-lapse with one acquisition per 2 s for both 568 nm and 491 nm illumination and played back at 10 frames per second, with time point 0 corresponding with early metaphase. The acquisition in the 568 nm channel at time point 0 shows localization of ePDZ::mCherry::GPR-1 in complete absence of blue light, as embryos were kept in the dark before image acquisition. Movie serves as a control to *Video 13*.
DOI: https://doi.org/10.7554/eLife.38198.030

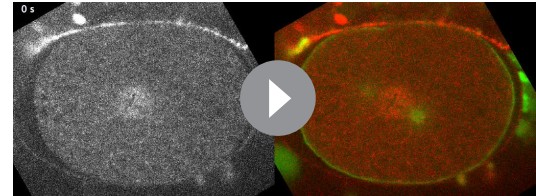

**Video 13.** Movie montage of a mitotic one-cell *C. elegans* embryo treated with *gpr-1/2* RNAi expressing endogenously labeled ePDZ::mCherry::DHC-1 (greyscale, left; red, right), the membrane anchor PH::eGFP::LOV and GFP::tubulin (both green, right). The movie shows relocalization of ePDZ::mCherry::DHC-1 to the cortex by global and continuous activation of cortical LOV using blue light. Images, which are single planes, were made as a time-lapse with one acquisition per 2 s for both 568 nm and 491 nm illumination and played back at 10 frames per second, with time point 0 corresponding with early metaphase. The acquisition in the 568 nm channel at time point 0 shows localization of ePDZ::mCherry::DHC-1 in complete absence of blue light, as embryos were kept in the dark before image acquisition. Movie corresponds to *Figure 6b*.
DOI: https://doi.org/10.7554/eLife.38198.031

indicate that dynein activation at microtubule plus ends requires the LIN-5/NuMA adaptor protein, similar to the requirement for an activating dynein adaptor in cargo transport (*McKenney et al., 2014*; *Schlager et al., 2014*; *Zhang et al., 2017*).

It is tempting to speculate that activation by LIN-5[NuMA] specifically promotes dynein function in generating membrane associated pulling forces. In the cytoplasm, dynein adopts an inactive conformation and does not show processive movement in the absence of an activating adaptor (*Reck-Peterson et al., 2018*). Adaptor proteins such as BicD2 and Hook3 interact with specific cargo as well as with the dynein tail regions and the universal activating complex dynactin (*McKenney et al., 2014*; *Schlager et al., 2014*; *Zhang et al., 2017*). Thereby, these adaptors promote the formation of stable and active dynein–dynactin-adaptor complexes. Recent stuctural analyses revealed that the coiled-coil regions of the adaptors extend in between dynein and dynactin, with multiple adaptor-specific interaction sites, and with several adaptors promoting the recruitment of two dynein complexes in parallel (*Grotjahn et al., 2018*; *Sladewski et al., 2018*; *Urnavicius et al., 2018*).

Although structural studies have focused on factors promoting minus-end directed transport, it is likely that dynein-dependent cortical pulling forces also involve an activating adaptor. As such, LIN-5[NuMA] could promote the assembly of stable dynein–dynactin complexes and release dynein autoinhibition. It is remarkable, however, that direct recruitment of dynein to the cortex did not result in cortical pulling forces, despite the co-recruitment of LIN-5 and the dynactin subunit DNC-1[p150]. At present, we do not know whether this reflects individual dynein–LIN-5 and dynein–dynactin interactions, or the formation of stable dynein–dynactin–LIN-5 complexes. The lack of pulling forces, despite co-recruitment of LIN-5[NuMA] and DNC-1[p150] together with dynein, may indicate that the complex needs to be anchored through LIN-5 in order to achieve a specific conformation and facilitate microtubule end-on pulling forces. Such a requirement could prevent inappropriate activation of dynein–dynactin–LIN-5[NuMA] complexes in the cytoplasm. In the optogenetic experiments in human carcinoma cells, dynein also localized p150 at the membrane, however in contrast to our findings, NuMA was not observed to be co-recruited in this study (*Okumura et al., 2018*). While the reason for this discrepancy is currently unknown, it might be related to the remarkably different kinetics of the two systems. Using the TULIP system, we observed membrane localization of ePDZ-tagged endogenous proteins within seconds, whereas in the iLID experiments in human carcinoma cells

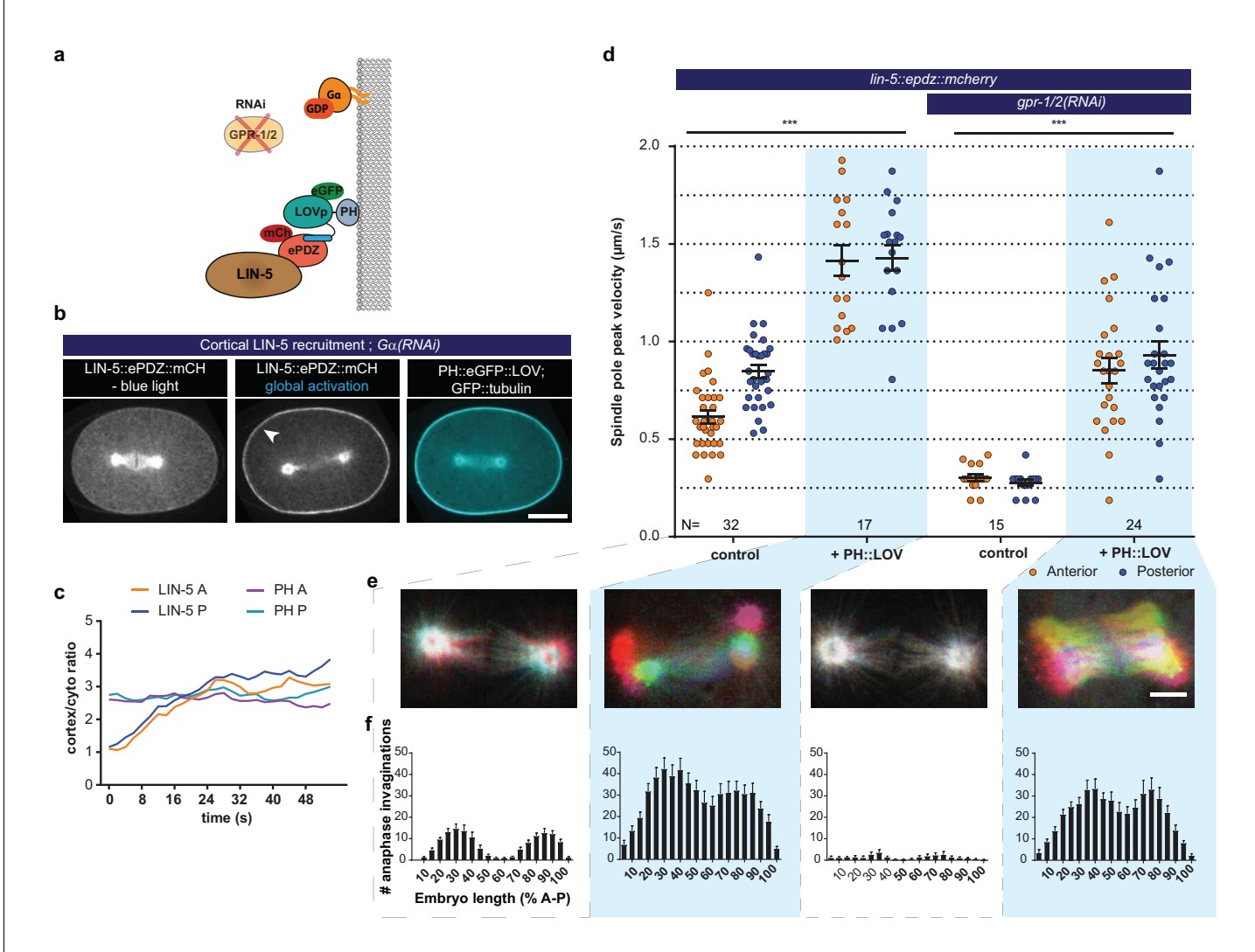

**Figure 7.** LIN-5 is a strong and essential activator of dynein-dependent cortical pulling forces. (**a**) Cartoon model illustrating the experiments in which LIN-5 is recruited to the cortex independently of the wild type Gα–GPR-1/2 anchor. (**b**) Spinning disk confocal images showing light-controlled recruitment of endogenous LIN-5 in the absence of Gα (arrow head). See also *Videos 14* and *15*. (**c**) Cortical LIN-5 recruitment during continuous activation of the ePDZ–LOV interaction, represented as the ratio of cortical/cytoplasmic signal. Scale bar: 10 μm. (**d**) Spindle severing experiments in combination with cortical recruitment of endogenous LIN-5. Control is the *lin-5::epdz::mcherry; gfp::tubulin* strain. Experimental conditions: combinations with *ph::egfp::lov* and *gpr-1/2(RNAi)*. Error bars: s.e.m. Welch's Student's t-test and Mann Whitney U test; ***p<0.001. (**e**) Maximum projections of spindle movements for 10 s using a temporal color coding scheme to visualize spindle movement as in *Figure 3c*. Scale bar: 5 μm. (**f**) Anaphase membrane invaginations plotted along anterior-posterior embryo length. Conditions were the same as for the connected experiments in d and e, except for the control, which was the *ph::egfp::lov; gfp::tubulin* strain and not *lin-5::epdz::mcherry*. Blue fields indicate conditions in which ePDZ and LOV components are present, and an ePDZ–LOV interaction is induced using blue light. See *Supplementary file 1* for detailed genotypes. Anterior is to the left in all microscopy images.

DOI: https://doi.org/10.7554/eLife.38198.032

The following figure supplements are available for figure 7:

**Figure supplement 1.** Cortical LIN-5 recruitment strongly increases cortical pulling force generation.
DOI: https://doi.org/10.7554/eLife.38198.033

**Figure supplement 2.** Cortical LIN-5 recruitment localizes dynein (DHC-1).
DOI: https://doi.org/10.7554/eLife.38198.034

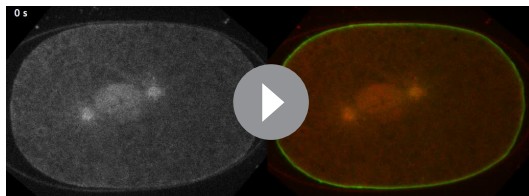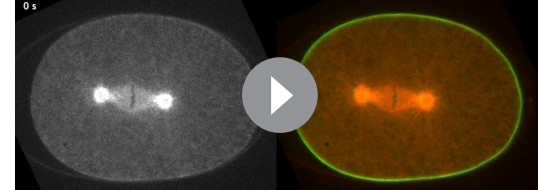

**Video 14.** Movie montage of a mitotic one-cell *C. elegans* embryo expressing endogenously labeled LIN-5::ePDZ::mCherry (greyscale, left; red, right) and the membrane anchor PH::eGFP::LOV and GFP::tubulin (both green, right). The movie shows relocalization of LIN-5::ePDZ::mCherry to the cortex by global and continuous activation of cortical LOV using blue light. Images, which are single planes, were made as a time-lapse with one acquisition per 2 s for both 568 nm and 491 nm illumination and played back at 10 frames per second, with time point 0 corresponding with metaphase. The acquisition in the 568 nm channel at time point 0 shows localization of LIN-5::ePDZ::mCherry in complete absence of blue light, as embryos were kept in the dark before image acquisition. Movie serves as a control to *Video 15*.
DOI: https://doi.org/10.7554/eLife.38198.035

**Video 15.** Movie montage of a mitotic one-cell *C. elegans* embryo treated with Gα RNAi expressing endogenously labeled LIN-5::ePDZ::mCherry (greyscale, left; red, right), the membrane anchor PH:: eGFP::LOV and GFP::tubulin (both green, right). The movie shows relocalization of LIN-5::ePDZ::mCherry to the cortex by global and continuous activation of cortical LOV using blue light. Images, which are single planes, were made as a time-lapse with one acquisition per 2 s for both 568 nm and 491 nm illumination and played back at 10 frames per second, with time point 0 corresponding with metaphase. The acquisition in the 568 nm channel at time point 0 shows localization of LIN-5::ePDZ::mCherry in complete absence of blue light, as embryos were kept in the dark before image acquisition. Movie corresponds to *Figure 7b*.
DOI: https://doi.org/10.7554/eLife.38198.036

accumulation took place over multiple minutes. It is possible that in the longer time frame, cytoplasmic dynein complexes are recruited to the cortex with adaptors other than NuMA.

Since only LIN-5 is strictly required for cortical pulling force generation, the question arises why a tripartite dynein anchor is conserved from worm to man. In yeast, dynein is localized by the single-component cortical anchor Num1, a coiled-coil domain protein with a PH-domain for membrane localization (*Ananthanarayanan, 2016*). Our ectopic ePDZ–LOV heterodimerization experiments show that membrane-tethered LIN-5 could suffice as a dynein anchor and activator, and that local regulation is needed to rotate and displace the spindle. Conceivably, the trimeric dynein anchor/ adaptor evolved in metazoans to augment context-specific regulation and reduce stochastic activation of spindle pulling forces.

Which factors may normally control the Gα–GPR-1/2 membrane anchor? The LET-99 protein was previously reported to restrict the localization of GPR-1/2 in a posterior-lateral band region of the one-cell embryo, thereby contributing to higher net pulling forces in the posterior direction (*Tsou et al., 2002*; *Tsou et al., 2003*). Our observation that direct membrane recruitment of GPR-1 overcomes this regulation is in agreement with LET-99 normally antagonizing Gα•GDP–GPR-1/2 interaction. Another level of Gα–GPR-1/2 regulation that remains incompletely understood is the contribution of RIC-8 in spindle positioning. The *ric-8* gene was discovered through 'resistant to inhibitors of cholinesterase 8' mutants in *C. elegans,* which are defective in Gα$_q$-stimulated neurotransmitter release (*Miller et al., 2000*). The discovery that *ric-8* acts also with GOA-1 in spindle positioning indicated a general role for RIC-8 in Gα regulation (*Miller and Rand, 2000*). Indeed, two different general functions have been reported. Based on in vitro experiments, mammalian RIC-8A acts as a non-receptor GEF, which surprisingly does not activate Gαβγ trimers, but shows higher affinity for free Gα•GDP and the Gα•GDP–LGN–NuMA complex (*Tall and Gilman, 2005*; *Tall et al., 2003*). In contrast, experiments in *Drosophila* and mouse embryonic stem cells demonstrated a chaperone function required for the cortical localization of Gα subunits (*David et al., 2005*; *Gabay et al., 2011*; *Hampoelz et al., 2005*; *Wang et al., 2005*). Only the *Drosophila* Gα$_i$ protein forms a complex with Pins–Mud (*Schaefer et al., 2001*), and *Drosophila* Gα$_i$, but not Gα$_o$, depends on RIC-8 for its cortical localization (*David et al., 2005*). Seemingly unifying these independently

described RIC-8 functions, *C. elegans* RIC-8 was shown to exhibit both GEF and chaperone activity, depending on the Gα subunit (*Afshar et al., 2004*; *Afshar et al., 2005*).

The *C. elegans* GOA-1 and GPA-16 Gα proteins act in a substantially redundant manner in spindle positioning, but diverge in other ways. GPA-16 is closest to the Gα$_i$ class, has been implicated only in spindle positioning, and depends on RIC-8 for its cortical localization (*Afshar et al., 2005*; *Bergmann et al., 2003*; *Gotta and Ahringer, 2001*). Thus, both *C. elegans* and *Drosophila* RIC-8 appears to act as a chaperone for Gα$_i$ to promote spindle positioning. Also similar to *Drosophila*, the Gα$_o$ GOA-1 subunit does not require RIC-8 for its membrane localization (*David et al., 2005*; *Afshar et al., 2005*). Instead, RIC-8 was reported to act as a GEF for GOA-1 (*Afshar et al., 2004*), and is usually considered to act as a GEF for Gα$_q$ and Gα$_o$ in neurotransmitter release. However, the contribution of a Gα$_o$ GEF in spindle positioning would mean that either Gα•GTP or Gα•GDP/GTP cycling promotes spindle pulling forces. Both of these possibilities seem unlikely in light of the results described here. The fact that Gα can be replaced with a PH–membrane anchor dismisses a general requirement for Gα•GTP in pulling force generation. Moreover, RIC-8 functioning in a Gα•GDP/GTP cycle is not supported by our knockout and membrane localization studies. Therefore, there is reason to question whether RIC-8 really functions as a GEF. The strongest support for such a function has come from in vitro experiments, in which RIC-8 showed rather inefficient GEF activity towards Gα•GDP (*Kant et al., 2016*; *Thomas et al., 2008*). It appears conceivable that incubation of Gα with a chaperone that affects its folding causes nucleotide dissociation. If correct, the membrane localized GOA-1 in *ric-8* mutants would reflect an incompletely functional form.

Despite the observed replaceability of Gα with a general membrane anchor, Gα•GTP has been reported to affect the spindle orientation in specific tissues (*Katanaev et al., 2005*; *Schaefer et al., 2001*; *Yoshiura et al., 2012*). In *Drosophila* neuroblasts and sensory organ precursor cells, canonical G-protein signaling is used to align cellular polarity with tissue polarity (*Katanaev et al., 2005*; *Yoshiura et al., 2012*). As such, Gα•GTP may indirectly contribute to spindle positioning. In addition, one of the *Drosophila* Pins GoLoco domains was found to preferentially interact with Gα$_o$•GTP, thereby linking the spindle positioning machinery with canonical G-protein signaling (*Kopein and Katanaev, 2009*; *Yoshiura et al., 2012*). Thus, while not needed for the generation of dynein-dependent cortical pulling forces, further studies will need to reveal to what extent Gα•GTP contributes to spindle positioning in a tissue or developmental context specific manner.

For our in vivo dissection of spindle positioning, we developed and applied methods for germline-specific gene knockout, tagging of endogenous proteins, reliable expression of foreign sequences in the germline, and light-inducible protein heterodimerization. These methods further expand the molecular biology toolbox for in vivo studies and can be broadly applied to other biological processes. Of particular interest is the acquired possibility to experimentally control the position of the spindle, for instance for future studies aimed at deciphering how the spindle determines the plane of cell cleavage, and whether specific cell-cell contacts affect cell fate.

## Materials and methods

### Key resources table

| Reagent type (species) or resource | Designation | Source or reference | Identifiers | Additional information |
|---|---|---|---|---|
| strain, strain background (*Caenorhabditis elegans*) | All strains derived from N2 | CGC | | |
| Antibody | mouse polyclonal anti-LIN-5 | *Lorson et al., 2000* | | |
| Antibody | mouse monoclonal anti-FLAG M2 | Sigma-Aldrich | Cat. No. F1804 | |
| Antibody | rabbit polyclonal anti-DHC-1 | *Gonczy et al., 1999* | | |
| software, algorithm | GLO (germline optimized) sequence optimization algorithm | this study | | accessible via http://104.131.81.59/ |

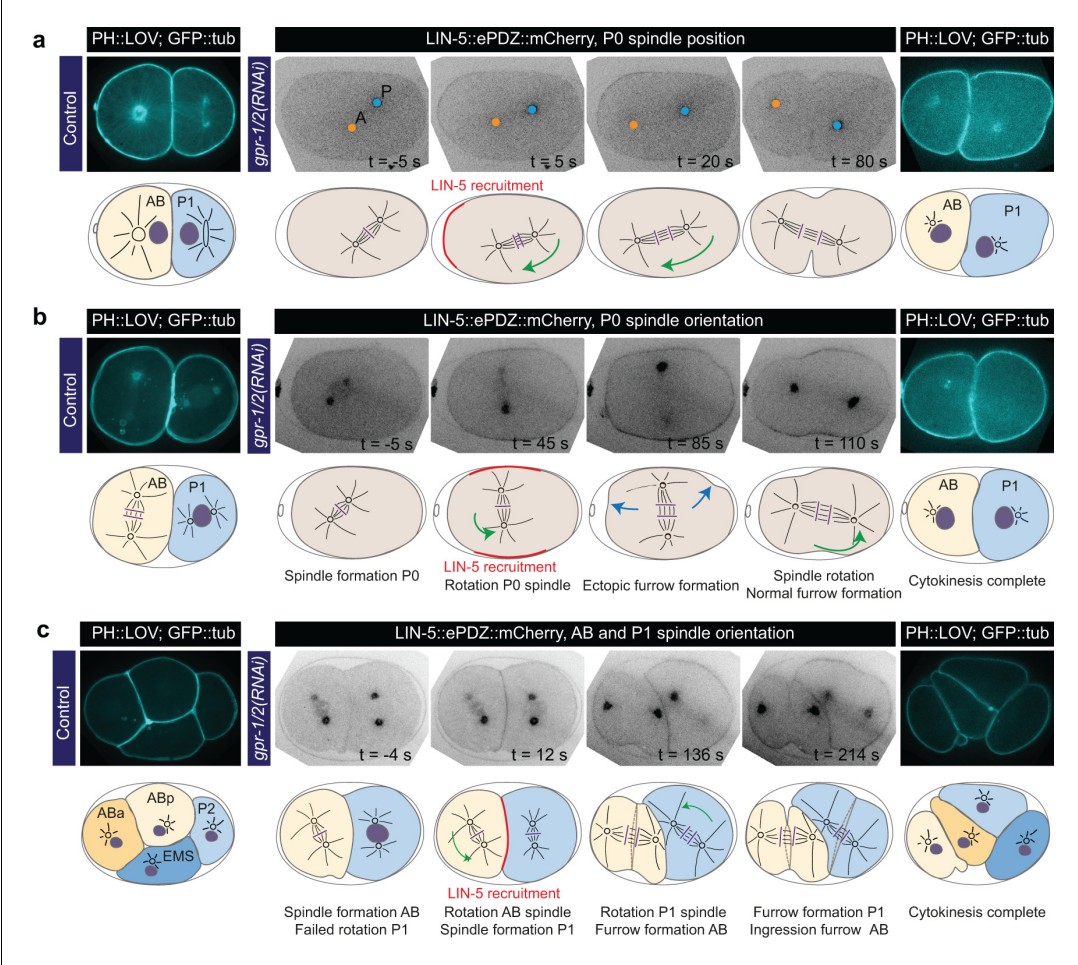

**Figure 8.** Experimentally induced spindle positioning by controlled localization of endogenous LIN-5. (**a**) Selected time points of *Video 18* showing induced anterior displacement of the P0 spindle upon local cortical recruitment of LIN-5. Images are annotated with centrosome positions shown as circles (orange, anterior pole; blue, posterior pole). (**b**) Selected time points of *Video 19* showing induced transverse P0 spindle orientation upon local cortical recruitment of LIN-5. Blue arrows, ectopic furrowing. (**c**) Selected time points of *Video 20* showing induced AB and P1 spindle rotation upon local cortical recruitment of LIN-5. In a, b, and c panels 1–4 show LIN-5::ePDZ::mCherry fluorescence, panel five shows PH::LOV and GFP::tubulin. Cartoons accompanying images illustrate key events. Red, local LIN-5 recruitment. Green arrows, spindle movements. leftmost panels show control two- and four-cell embryos labeled with PH::eGFP::LOV and GFP::tubulin. See Materials and methods for the local activation procedure. See *Supplementary file 1* for detailed genotypes. Anterior is to the left in all microscopy images.

DOI: https://doi.org/10.7554/eLife.38198.039

## *C. elegans* strains and maintenance

The names and associated genotypes of *C. elegans* strains used in this study are included in *Supplementary file 1*. Animals were maintained at either 15 or 20°C as described previously (*Brenner, 1974*). Strains expressing both ePDZ and LOV protein motifs were regarded as light-sensitive and thus cultured in the dark, and transferred while using red light only. Animals were kept on plates that contained nematode growth medium (NGM) that had been seeded with OP50 *Escherichia coli* bacteria.

## Molecular cloning

DNA vector-based repair templates to be used for CRISPR/Cas9-mediated genome editing were designed in A plasmid Editor (M. Wayne Davis) to include 500–1500 bp homology arms. These and all other sequences used were generated starting from either purified *C. elegans* genomic DNA or pre-existing vectors via PCR amplification using Q5 Hot Start High-Fidelity DNA Polymerase (New

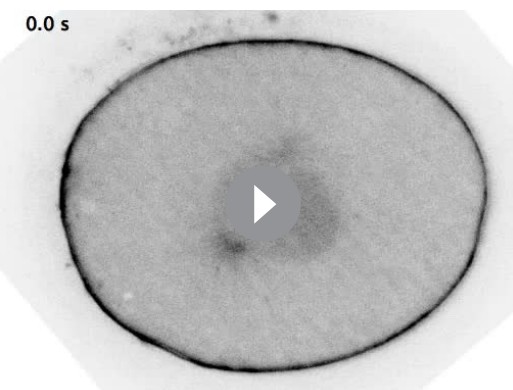

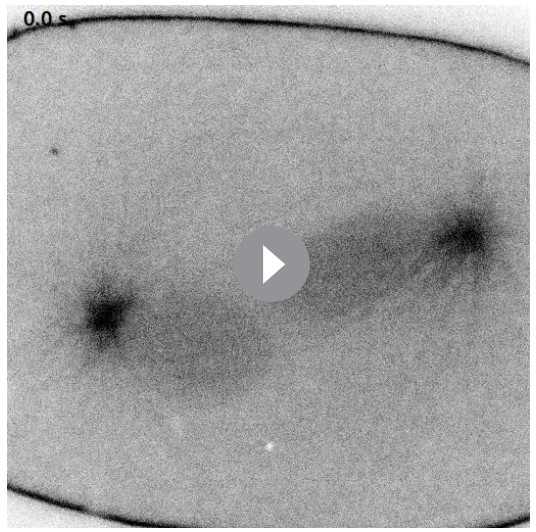

**Video 16.** Movie montage of a mitotic one-cell *C. elegans* embryo treated with *gpr-1/2* RNAi expressing endogenously labeled LIN-5::ePDZ::mCherry (not shown), the membrane anchor PH::eGFP::LOV and GFP::tubulin (both inverted greyscale). The movie shows excessive rocking of centrosomes with associated pronuclei prior to mitotic spindle assembly. Images, which are single planes, were made as a streaming acquisition with 0.5 s of exposure and played back at 20 frames per second, with time point 0 corresponding with late prophase. Movie does not correspondto a figure, but is discussed in the text.
DOI: https://doi.org/10.7554/eLife.38198.037

**Video 17.** Movie montage of a mitotic one-cell *C. elegans* embryo expressing endogenously labeled LIN-5::ePDZ::mCherry (not shown) and the membrane anchor PH::eGFP::LOV (inverted greyscale). The movie shows separation of centrosomes and associated pronuclei in prophase upon global and continuous activation of LOV with blue light. Images, which are single planes, were made as a streaming acquisition with 0.5 s of exposure and played back at 10 frames per second, with time point 0 corresponding with prophase. Movie does not correspond to a figure, but is discussed in the text.
DOI: https://doi.org/10.7554/eLife.38198.038

England Biolabs). A list of all cloning, repair template and genotyping primers (Integrated DNA technologies) and DNA templates used has been included in *Supplementary file 2*. PCR fragments were gel purified (Qiagen), their concentrations measured using a BioPhotometer D30 (Eppendorf) and then ligated into pBSK by Gibson assembly (New England Biolabs). gRNA vectors were generated by annealing of antisense oligonucleotide pairs and subsequent ligation into BbsI-linearized pJJR50 or BsaI-linearized pMB70 using T4 ligase (New England Biolabs). All DNA vectors used for genome editing were transformed into DH5α competent cells and subsequently purified by midi-prep (Qiagen).

## Design of germline-optimized coding sequences

Custom Perl scripts were written to design germline-optimized coding sequences according to the algorithm described in the legend of *Figure 2—figure supplement 1*. After designing each coding sequence, we inserted either (1) normal synthetic introns with the sequence gtaagttt(n36)ttttcag, where n36 is a 36 bp random DNA sequence with 30% GC content; or (2) PATC introns (*Frøkjær-Jensen et al., 2016*). Our design algorithm is accessible via a web interface at http://104.131.81.59/, and the source code can be found at https://github.com/dannyhmg/germline; copy archived at https://github.com/elifesciences-publications/germline. Germline-optimized sequences were synthesized as gBlocks (Integrated DNA Technologies) and single-copy transgenes were generated using standard methods (*Frøkjær-Jensen et al., 2012*). Please refer to *Supplementary file 2* for detailed sequence features of each transgene.

## Design of inducible germline-specific gene knockout

*loxP* and *loxN* sequences were integrated in the endogenous loci of essential genes (see CRISPR/Cas9-mediated genome editing section for details). For the FLP recombinase, the hyperactive FLP G5D variant (*Schwartz and Jorgensen, 2016*) was used (pMLS262; Addgene #73718). For germline-

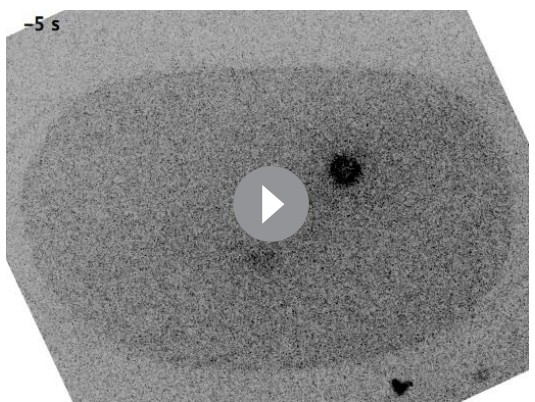
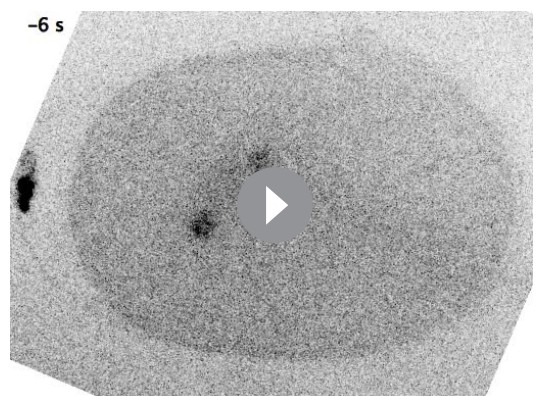

**Video 18.** Movie montage of a mitotic one-cell *C. elegans* embryo treated with *gpr-1/2* RNAi expressing endogenously labeled LIN-5::ePDZ::mCherry (inverted greyscale) and the membrane anchor PH::eGFP::LOV (not shown). The movie shows anterior displacement of the spindle and subsequent inverted asymmetric division resulting in a small anterior and large posterior blastomere after local recruitment of LIN-5::ePDZ::mCherry to the anterior cortex. Images, which are single planes, were made as a time-lapse with one acquisition per 5 s and played back at five frames per second, with time point 0 corresponding with metaphase. Movie corresponds to *Figure 8a*.
DOI: https://doi.org/10.7554/eLife.38198.040

**Video 19.** Movie montage of a mitotic one-cell *C. elegans* embryo treated with *gpr-1/2* RNAi expressing endogenously labeled LIN-5::ePDZ::mCherry (inverted greyscale) and the membrane anchor PH::eGFP::LOV (not shown). The movie shows artificial transverse positioning of the metaphase mitotic spindle and its subsequent correction to an anterior-posterior position in late anaphase after local recruitment of LIN-5::ePDZ::mCherry to the opposing equatorial cortexes. Images, which are averages of groups of 2 subsequent frames, were made as a time-lapse with one acquisition per 2 s and played back at five frames per second, with time point 0 corresponding with metaphase. Movie corresponds to *Figure 8b*.
DOI: https://doi.org/10.7554/eLife.38198.041

specific expression, we used a long version of the *pie-1* promoter including enhancer sequences (pAZ132, a kind gift from A. A. Hyman). To initiate the recombination cascade, the germline-specific FLP expression vector was injected in P0 mothers with the following protocol to favor germline expression (Personal communication Oliver Hobert, Worm Breeders Gazette, February 21, 2013); linearized FLP construct (2 ng/μl), PvuII digested *E. coli* genomic DNA (150 ng/μl), co-injection marker *Pmyo-2::tdtomato* (2 ng/μl). Transgenic F1 animals were singled and allowed to lay eggs for at least 24 hr. From F1 with 100% embryonic lethal broods (*ric-8* and *rgs-7* are essential for embryogenesis), early embryos were isolated and used for spindle severing experiments.

## CRISPR/Cas9-mediated genome editing

Either the wild type N2 or SV1818 (*pha-1*(*e2123*ts) 4x outcrossed) *C. elegans* genetic background was used for the generation of CRISPR/Cas9 alleles. Injection mixes with a total volume of 50 μl were prepared in MilliQ HR$_2$RO and contained a combination of 50 ng/μl *Peft-3:cas9* (Addgene ID #46168 (*Friedland et al., 2013*) or 60 ng/μl pJW1285 (Addgene ID #61252 [*Dickinson et al., 2013*]), 50–100 ng/μl *u6::sgRNA* (targeting genomic sequences listed in *Supplementary file 2*), 50 ng/μl of (PAGE-purified oligonucleotide) repair template and 2.5 ng/μl of the co-injection pharyngeal marker *myo-2p::tdtomato*. Injection mixes were spun down in a microcentrifuge (Eppendorf) for at least 10 min at 13,000 RPM prior to use. Young adult hermaphrodites were injected in the germline using an inverted micro-injection setup (Eppendorf). After injection, animals were singled and grown at 15 or 20°C. F1 animals were then picked to a total of at least 96, and grown with two or three animals per plate for 7–8 days at 20°C until freshly starved. Half a plate containing F2 and F3 animals was then washed off with M9 medium supplemented with 0.05% Tween-20, and subsequently lysed to extract genomic DNA. Some knock-ins were obtained using co-CRISPR selection: rescue of *pha-1*(*e2123*ts) (*Ward, 2015*), generation of visible *unc-22* (*Kim et al., 2014*) or *dpy-10* (*Paix et al., 2015*) phenotypes, or integration of an self-excisable cassette carrying a visible marker (*Dickinson et al., 2015*). Genotyping was carried out by PCR amplification with OneTaq polymerase (New England Biolabs)

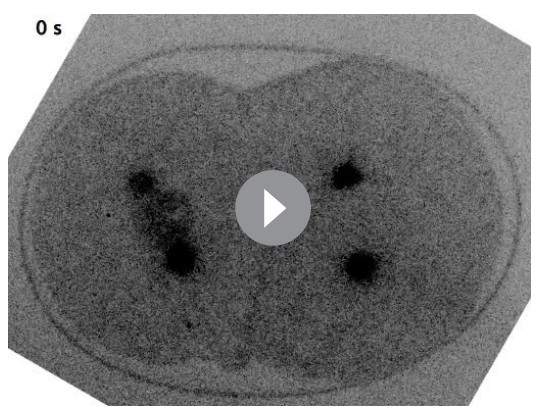

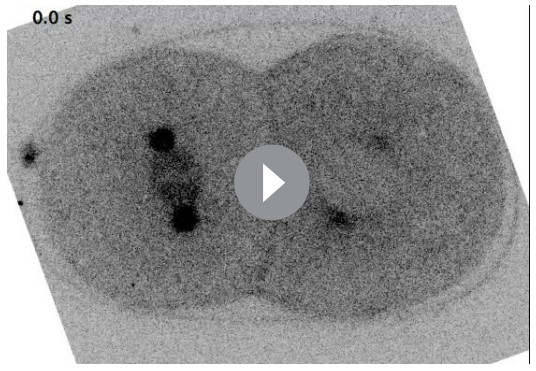

**Video 20.** Movie montage of a mitotic two-cell *C. elegans* embryo treated with *gpr-1/2* RNAi expressing endogenously labeled LIN-5::ePDZ::mCherry (inverted greyscale) and the membrane anchor PH::eGFP::LOV (not shown). The movie shows artificial rotation of transverse aligned AB and P1 spindles to an anterior-posterior position, and concurrent reorientation of the cleavage planes after local recruitment of LIN-5::ePDZ::mCherry to the central region where AB and P1 cortexes touch. Images, which are averages of groups of 2 subsequent frames, were made as a streaming acquisition with 0.5 s of exposure and played back at 10 frames per second, with time point 0 corresponding with metaphase in the AB blastomere. Movie corresponds to *Figure 8c*.
DOI: https://doi.org/10.7554/eLife.38198.042

**Video 21.** Movie montage of a mitotic two-cell *C. elegans* embryo treated with *gpr-1/2* RNAi expressing endogenously labeled LIN-5::ePDZ::mCherry (inverted greyscale) and the membrane anchor PH::eGFP::LOV (not shown). The movie shows artificial rotation of transverse aligned AB and P1 spindles to an anterior-posterior position, and concurrent reorientation of the cleavage planes after local recruitment of LIN-5::ePDZ::mCherry to the central region where AB and P1 cortexes touch. Images, which are averages of groups of 2 subsequent frames, were made as a streaming acquisition with 0.5 s of exposure and played back at 20 frames per second, with time point 0 corresponding with metaphase in the AB blastomere. Movie corresponds to *Figure 8c*.
DOI: https://doi.org/10.7554/eLife.38198.043

of genome sequences using primers annealing in the inserted sequence and a genomic region not included in the repair template. Confirmed alleles were subsequently sequenced (Macrogen Europe).

## Spinning disk microscopy

Prior to live imaging, embryos were dissected from adult hermaphrodites onto coverslips (Menzel-Gläser) in 0.8x egg salts buffer (94 mM NaCl, 32 mM KCl, 2.7 mM CaCl2, 2.7 mM MgCl2, 4 mM HEPES, pH 7.5; (*Tagawa et al., 2001*) or M9, and mounted on 4% agarose pads. Spinning disk imaging of embryos was performed using a Nikon Eclipse Ti with Perfect Focus System, Yokogawa CSU-X1-A1 spinning disk confocal head, Plan Apo VC 60x N.A. 1.40 oil and S Fluor 100x N.A. 0.5–1.3 (at 1.3, used for UV-laser photo-ablation) objectives, Photometrics Evolve 512 EMCCD camera, DV2 two-channel beam-splitter for simultaneous dual-color imaging, Vortran Stradus 405 nm (100 mW), Cobolt Calypso 491 nm (100 mW), Cobolt Jive 561 nm (100 mW), Vortran Stradus 642 nm (110 mW) and Teem Photonics 355 nm Q-switched pulsed lasers controlled with the ILas system (Roper Scientific France/PICT IBiSA, Institut Curie, used for photo-ablation), ET-GFP (49002), ET-mCherry (49008), ET-GFPmCherry (49022) and ET-Cy5 (49006) filters, ASI motorized stage MS-2000-XYZ with Piezo Top Plate, and Sutter LB10-3 filter wheel. The microscope was operated using MetaMorph 7.7 software and situated in a temperature-controlled room (20˚C). The temperature of the stage and objective was controlled at 25˚C with a Tokai Hit INUBG2E-ZILCS Stage Top Incubator during experiments. Images were acquired in either streaming mode with 250 or 500 ms exposure, or time-lapse mode with 250, 500 or 1500 ms exposure and 2 or 5 s intervals. Laser power and exposure times were kept constant within experiments. For the quantification of membrane invaginations embryos were imaged by 250 ms exposure stream acquisition starting in the DNA plane at anaphase onset, as judged by GFP::tubulin signal. During anaphase, the spinning disk imaging plane was moved as close to the membrane as possible while keeping the cytosol discernable from the membrane signal. Acquisitions were terminated at early telophase, as judged by the PH::eGFP::LOV signal. For

experiments involving balanced *epdz::mcherry::dhc-1/+*, each animal was confirmed to be positive for *epdz::mcherry::dhc-1* by fluorescence before the experiment. Images acquired by spinning disk microscopy were rotated, cropped, annotated, provided with scale bars, and processed further by linear adjustment of brightness and contrast using ImageJ and FIJI. Fluorophores used in this study include (e)GFP, mCherry, Alexa-488, Alexa-568 and Atto 647N.

## RNA-mediated interference (RNAi)

For immunohistochemistry experiments L4 hermaphrodites were grown on RNAi plates seeded with HT115 *Escherichia coli* bacteria strains generating double-stranded RNA (dsRNA) targeting genes of interest (*goa-1, gpa-16, gpr-1*) for 48 hr at 15℃ prior to fixation (*Timmons and Fire, 1998*). For all other gene knock-down experiments, young adult hermaphrodites were injected with dsRNA targeting genes of interest (*goa-1, gpa-16, gpr-1, ric-8, rgs-7*) and grown for 48 hr at 15℃ (*Fire et al., 1998*) prior to experiments. To generate dsRNA, coding regions of genes of interest were PCR amplified using Q5 Hot Start High-Fidelity DNA Polymerase (New England Biolabs). These PCR products were used as templates for in vitro dsRNA synthesis (MEGAscript T7 transcription kit, ThermoFisher Scientific). dsRNA was diluted 5x in DEPC $HR_2RO$ prior to micro-injection. ORF clones from the Vidal and Ahringer RNAi libraries were used (*Kamath et al., 2003*; *Rual et al., 2004*).

## Spindle severing assays

Mitotic spindle severing was performed in essence as described (*Grill et al., 2001*; *Portegijs et al., 2016*). One-cell embryos expressing GFP- or mCherry-labeled tubulin were imaged during mitosis using the spinning disk microscope setup described above, equipped with a Teem Photonics 355 nm Q-switched pulsed laser controlled with the ILas system (Roper Scientific France/PICT IBiSA, Institut Curie). At anaphase onset, as judged by spindle morphology and mobility, spindles were severed as shown in *Figure 1c* and *Video 2*. Centrosome displacement was recorded by 500 ms exposure streaming acquisition, and peak velocities were subsequently extrapolated using the FIJI TrackMate plugin.

## Dark state experiments and local recruitment of ePDZ-tagged proteins to membrane LOV

Dark state experiments were performed on the spinning disk setup described above. For local photoactivation of LOV2 in *C. elegans* embryos, light was applied in a region of variable size depending on each individual experiment using a 491 nm laser controlled with the ILas system (Roper Scientific France/PICT IBiSA, Institut Curie). Due to high sensitivity of LOV2 to blue light and variations in laser power, embryos of strain SV2061 (expressing diffuse ePDZ::mCherry and PH::eGFP::LOV) were used to calibrate the amount of laser power required for local activation of LOV2 prior to experiments. During both global and local photoactivation assays and dark state spindle severing experiments embryos were kept away from blue light as much as practically feasible. To this end, aluminum foil was used to cover the microscope setup, and optical filters were inserted in the light path to remove LOV2-activating wavelengths from the transmitted light used to locate embryos on slides. Prior to experimental use of embryos, unintended premature cortical recruitment of ePDZ-mCherry or ePDZ-mCherry-LIN-5 was assessed by observation of mCherry localization patterns.

## Antibodies and immunocytochemistry

For immunostaining of *C. elegans* embryos, embryos were dissected from adults in 10 µl MilliQ $HR_2RO$ on slides coated with poly-L-lysine. Samples were then freeze-cracked and fixed in methanol for 5 min. at −20℃ and subsequently in acetone for 5 min. at −20℃. Embryos were then rehydrated in phosphate buffered saline +0.05% Tween-20 (PBST), blocked for 1 hr at 4℃ in PBST +1% bovine serum albumin and 1% goat serum (Sigma-Aldrich), and then incubated at room temperature with primary antibodies for 1 hr and then with secondary antibodies for 45 min., both in blocking solution, with four 10 min washes in PBST following each antibody mix. Finally, embryos were embedded in ProLong Gold Antifade with DAPI. Primary antibodies used were mouse anti-LIN-5 [1:10, (*Lorson et al., 2000*), mouse anti-FLAG M2 (Sigma-Aldrich) and rabbit anti-DHC-1 (1:100 (*Gonczy et al., 1999*); a kind gift from P. Gönczy). Secondary antibodies used were goat anti-rabbit Alexa-488, goat anti-rabbit Alexa-568, goat anti-mouse Alexa-488, goat anti-mouse Alexa-568

(Invitrogen) and goat anti-mouse Atto 647N (Sigma-Aldrich), all at 1:500 dilution in blocking solution. Imaging of immunolabeled embryos was performed on the spinning disk setup described above.

## Data analysis: membrane invaginations and fluorescence intensity measurements

All quantitative spinning disk image analyses were performed in either ImageJ or FIJI. For quantification of membrane invaginations, movies were limited to the 200 frames (50 s) preceding the onset of telophase. Images were then cropped to include the outer limits of the PH::eGFP signal. Transient cortical dots were tracked manually using the MTrackJ ImageJ plugin. Larger, more static structures result from membrane ruffles, which are distinct from the more dynamic invaginations, as can be seen in *Video 11*. To yield the distribution of invaginations on the length axis of the visible embryo cortex, recorded x coordinates were incremented into groups of 5% embryo length each. To quantify the cortical recruitment and dynamics of ePDZ::mCherry, ePDZ::mCherry::GPR-1, LIN-5::ePDZ::mCherry and ePDZ::mCherry::DHC-1 by PH::eGFP::LOV, multiple 20 px wide linescans were drawn perpendicular to the membrane per analyzed embryo. An intensity profile was plotted per linescan at each acquired time point, from each of which an average of the maximum three pixel values was extracted to yield the peak intensity values at the membrane. Each intensity measurement was first corrected for background noise with a value measured outside of the embryo in a 50 × 50 px region of interest, and cortex to cytoplasm intensity ratios were calculated using average cytoplasmic intensity measurements in a 50 × 50 or 29 × 23 px region of interest at all timepoints analyzed. Fluorescence intensity measurements as measure for *Cre(FLPon)* activation (*Figure 1—figure supplement 2*) were taken as total embryo average intensity minus background signal using ImageJ measurement tool. The half time of ePDZ-LOV interaction after a pulse activation was inferred from a non-linear, single component regression. All numerical data processing and graph generation was performed using Excel 2011 (Microsoft) and Prism 7 (GraphPad software, inc.).

## Statistical analysis

All data were shown as means with SEM. Statistical significance as determined using two-tailed unpaired Student's t-tests, Mann-Whitney U tests and the Wilcoxon matched-pairs signed rank test. Correlation coefficients between two data sets were calculated using Pearson $r$ correlation tests or Spearman rank correlation tests. Data sets were assessed for their fit to a Gaussian distribution using the D'Agostino-Pearson omnibus K2 normality test prior to application of appropriate statistical test. A p-value of <0.05 was considered significant. *p<0.05; **p<0.01; ***p<0.001; ****p<0.0001. All statistical analyses were performed in Prism 7 (GraphPad software, inc.).

## Code availability

Our design algorithm is accessible via a web interface at http://104.131.81.59/, and the source code can be found at https://github.com/dannyhmg/germline; copy archived at https://github.com/elifesciences-publications/germline.

## Data availability

The data that support the findings of this study are included in the supplementary information.

## Acknowledgements

We thank S Jonis, Y Onderwater, H Pires, P van Bergeijk, P Gönczy, A Skop, and M Harterink for reagents and L Kapitein for technical advice. We also thank all the members of the van den Heuvel, Akhmanova, Goldstein, Boxem, and Kapitein groups for helpful discussion and general support. We further thank A Thomas for critically reading the manuscript. We acknowledge Wormbase and the Biology Imaging Center at the Faculty of Sciences, Department of Biology, Utrecht University. Some strains were provided by the Caenorhabditis Genetics Center (CGC), which is funded by NIH Office of Research Infrastructure Programs (P40OD010440).

## Additional information

### Competing interests

Anna Akhmanova: Deputy editor of eLife Magazine. The other authors declare that no competing interests exist.

### Funding

| Funder | Grant reference number | Author |
|---|---|---|
| Nederlandse Organisatie voor Wetenschappelijk Onderzoek | NWO TOP/ECHO grant 711.015.001 | Lars-Eric Fielmich Sander Van den Heuvel |
| National Institutes of Health | NIH R01 GM083071 | Bob Goldstein |
| Helen Hay Whitney Foundation | | Daniel J Dickinson |
| European Research Council | Synergy grant 609822 | Ruben Schmidt Anna Akhmanova |
| National Institutes of Health | NIH K99 GM115964 | Daniel J Dickinson |

The funders had no role in study design, data collection and interpretation, or the decision to submit the work for publication.

### Author contributions

Lars-Eric Fielmich, Conceptualization, Data curation, Formal analysis, Validation, Investigation, Visualization, Methodology, Writing—original draft, Writing—review and editing; Ruben Schmidt, Conceptualization, Data curation, Formal analysis, Validation, Investigation, Visualization, Methodology, Writing—review and editing; Daniel J Dickinson, Conceptualization, Resources, Data curation, Software, Formal analysis, Funding acquisition, Validation, Investigation, Visualization, Methodology, Writing—original draft, Writing—review and editing; Bob Goldstein, Conceptualization, Resources, Formal analysis, Supervision, Funding acquisition, Methodology, Project administration; Anna Akhmanova, Conceptualization, Resources, Supervision, Funding acquisition, Project administration, Writing—review and editing; Sander van den Heuvel, Conceptualization, Resources, Formal analysis, Supervision, Funding acquisition, Methodology, Writing—original draft, Project administration, Writing—review and editing

### Author ORCIDs

Lars-Eric Fielmich http://orcid.org/0000-0003-0247-1298
Ruben Schmidt http://orcid.org/0000-0001-9187-5424
Daniel J Dickinson http://orcid.org/0000-0003-2651-2584
Bob Goldstein http://orcid.org/0000-0001-6961-675X
Anna Akhmanova http://orcid.org/0000-0002-9048-8614
Sander van den Heuvel http://orcid.org/0000-0001-9015-7463

### Decision letter and Author response

Decision letter https://doi.org/10.7554/eLife.38198.053
Author response https://doi.org/10.7554/eLife.38198.054

## Additional files

### Supplementary files

• Supplementary file 1. Table of all *C. elegans* genotypes used in this study, listed per figure.
DOI: https://doi.org/10.7554/eLife.38198.044

• Supplementary file 2. Table of all oligonucleotides used in this study, listed per transgene.
DOI: https://doi.org/10.7554/eLife.38198.045

• Supplementary file 3. Table of the raw data points generated in this study, listed per figure.
DOI: https://doi.org/10.7554/eLife.38198.046

• Transparent reporting form
DOI: https://doi.org/10.7554/eLife.38198.047

## Data availability

Our design algorithm is accessible via a web interface at http://104.131.81.59/, and the source code can be found at https://github.com/dannyhmg/germline (copy archived at https://github.com/elifes-ciences-publications/germline). The data that support the findings of this study are included in the supplementary information. All key plasmids and strains will be deposited with Addgene and the CGC upon publication.

The following previously published datasets were used:

| Author(s) | Year | Dataset title | Dataset URL | Database and Identifier |
|---|---|---|---|---|
| Hillier L, Reinke V, Green P, Hirst M, Marra MA, Waterston RH | 2008 | Massively parallel sequencing of the polyadenylated transcriptome of C. elegans | https://trace.ncbi.nlm.nih.gov/Traces/sra/sra.cgi?study=SRP000401 | NCBI Sequence Read Archive, SRP000401 |
| Gerstein MB, Rozowsky J, Yan KK, Wang D, Cheng C, Brown JB, Davis CA, Hillier L, Sisu C, Li JJ | 2014 | Comparative analysis of the transcriptome across distant species | https://www.encodeproject.org/publication-data/ENCSR145VDW/ | ENCODE, ENCSR145VDW |

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
