## [Decision Letter]

Thank you for submitting your article "Optogenetic dissection of mitotic spindle positioning in vivo" for consideration by *eLife*. Your article has been reviewed by Andrea Musacchio as the Senior Editor, a Reviewing Editor, and three reviewers. The reviewers have opted to remain anonymous.

The reviewers have discussed the reviews with one another and the Reviewing Editor has drafted this decision to help you prepare a revised submission.

Summary:

Fielmich et al., present an elegant optogenetic analysis that demonstrates the sufficiency of light-induced cortical localization of LIN-5/NuMa to activate dynein-dependent cortical pulling forces that orient mitotic spindles in the early *C. elegans* embryo. Importantly, they show that light-induced cortical localization is sufficient to induce increased pulling forces in the absence of GPR-1,2/LGN that normally acts to tether LIN-5 to the membrane via an interaction with Gα∙GDP, and that light-induced cortical localization of dynein alone is not sufficient to generate increased pulling forces. This is an impressive use of optogenetics to establish the sufficiency of cortical LIN-5 to activate dynein in the absence of upstream regulation. The authors provide thorough and appropriate statistical analyses indicating the significance of their findings and present a thorough and compelling genetic analysis of this issue. The authors also introduce a novel approach to avoiding germline silencing of transgenes in *C. elegans* that will be of substantial interest to other researchers studying processes controlled by germline expressed gene products. Similarly, their use of germline Cre-Lox gene deletion to bypass earlier essential requirements and achieve a more substantial reduction in gene function compared to RNAi is rigorous and impressive.

Essential revisions:

1) Direct targeting of dynein heavy chain to the cortex also recruits LIN-5 to the cortex (Figure 6—figure supplement 2). Given that LIN-5 is proposed to act as a dynein activator, it is surprising that no cortical pulling forces are generated in this condition. One possibility is that dynactin is missing from the cortical dynein/LIN-5 assembly when dynein is directly targeted to the cortex. The authors should perform IF against a dynactin component to address this.

2) An interesting analysis of the manuscript is the dissection of the Gα∙GTP-cycle and its regulation by RGS-7 and RIC-8. The authors generically refer to Gα to indicate both GOA-1 and GPA-16. It is known that in *Drosophila* neuroblasts two Gα subunits synergies to drive spindle orientation (Yoshiura, 2012). To understand whether also in *C. elegans* the two Gα subunits play distinct roles, it would be interesting to explore the phenotypic effect of ablating of the Gα subunits individually when targeting RGS-7 or RIC-8 to the cortex. Alternatively, the possibility of the two Gα subunits playing distinct roles should be discussed.

3) The authors report that membrane invaginations appear upon GPR-1 ectopic targeting, which is an intriguing observation. Do the authors think that these invaginations are physiological or that they are generated by increased cortical levels of GPR-1? What are the levels of GPR-1 ectopically accumulated at the membrane by the TUPLIP photo-inducible system compared to the levels recruited by Gα? It would be important to discuss whether these invaginations might be a common feature occurring in cellular systems other than *C. elegans* embryo as a general mechanism of force generators' activation.

4) The authors find that Gα knockdown results in a shift in the band of decreased pulling forces that in wild-type zygotes depends on LET-99 for its slightly posteriorized location. While this is a minor issue, it is somewhat puzzling. Have the authors, or others in previous studies, examined LET-99 localization in Gα mutants? If this has been done, it would be worth mentioning in the Discussion section; if not perhaps the authors could look at LET-99 localization and see if it is shifted similarly?

5) The authors nicely count spots associated with membrane invaginations as an alternative to pulling forces to quantify the effects of light-induced cortical localization of these regulators. In the images of the spots (Figure 5G), there seem to be spots at the very posterior and anterior poles, and yet the bar graphs show an absence of such foci at the termini. Why are the numbers reduced at the extreme ends, and if they are, why do there seem to be quite a few of them, particularly at the anterior pole, in the lower image of Figure 5G? I can imagine where it is largely a surface area issue, but some clarification on this point might be worthwhile.

6) The authors never present any data in which LIN-5 is knocked down using RNAi. In particular, it would seem like a worthwhile thing to do when documenting the light-induced cortical localization of Dynein (presumably it would be independent of not only GPR-1/2, as shown, but also of LIN-5). Perhaps more importantly, as something of a control, it might be worth showing that the effect of light induced cortical localization of LIN-5 is in fact LIN-5 dependent (loss of pulling forces after blue light exposure in lin-5(RNAi) zygotes). This is a minor concern, given the other controls. But I am curious as to why the authors have not used any lin-5(RNAi) in their analysis at some point, just for the sake of completeness if nothing else.

7) While the authors point out how their analysis goes beyond two previous studies that also addressed the role of LIN-5/NuMa in activating cortical dynein pulling forces (Kota et al., 2012; Ségelin et al., 2010), the authors need to address in similar fashion results just published in *eLife* from Okumura et al., 2018) concerning vertebrate NuMa. The authors should cite this paper and discuss their results in the context of the ones reported by the Kiyomitsu paper.

8) Unless I missed it, the authors do not describe in the text, figure legend, or methods exactly how they did the localized light-induced increases in cortical LIN-5. Some mention of the control of blue light illumination used for restricted localization should be provided in the Materials and methods section at least.

---

## [Author Response]

Summary:Fielmich et al., present an elegant optogenetic analysis that demonstrates the sufficiency of light-induced cortical localization of LIN-5/NuMa to activate dynein-dependent cortical pulling forces that orient mitotic spindles in the early C. elegans embryo. Importantly, they show that light-induced cortical localization is sufficient to induce increased pulling forces in the absence of GPR-1,2/LGN that normally acts to tether LIN-5 to the membrane via an interaction with Gα∙GDP, and that light-induced cortical localization of dynein alone is not sufficient to generate increased pulling forces. This is an impressive use of optogenetics to establish the sufficiency of cortical LIN-5 to activate dynein in the absence of upstream regulation. The authors provide thorough and appropriate statistical analyses indicating the significance of their findings and present a thorough and compelling genetic analysis of this issue. The authors also introduce a novel approach to avoiding germline silencing of transgenes in C. elegans that will be of substantial interest to other researchers studying processes controlled by germline expressed gene products. Similarly, their use of germline Cre-Lox gene deletion to bypass earlier essential requirements and achieve a more substantial reduction in gene function compared to RNAi is rigorous and impressive.Essential revisions:1) Direct targeting of dynein heavy chain to the cortex also recruits LIN-5 to the cortex (Figure 6—figure supplement 2). Given that LIN-5 is proposed to act as a dynein activator, it is surprising that no cortical pulling forces are generated in this condition. One possibility is that dynactin is missing from the cortical dynein/LIN-5 assembly when dynein is directly targeted to the cortex. The authors should perform IF against a dynactin component to address this.

We agree that this is an important question and have performed the requested experiments. We made use of two independent antibodies to detect the DNC-1^p150(Glued)^ dynactin component: an anti-DNC-1 antibody (Skop and White, 1998) to recognize endogenous DNC-1^p150^, and an anti-FLAG antibody to detect endogenously tagged DNC-1::mNG::3xFLAG (Strain generated by Goldstein lab, 2018). Immunohistochemistry experiments with the two antibodies produced very similar results and demonstrate that DNC-1^p150^ is readily co-recruited to the cortex when ePDZ::DHC-1 is targeted to PH::LOV.

These data are included in the text (subsection “Cortical LIN-5MuD/NuMA is essential and sufficient for dynein-dependent pulling force generation”) and Figure 6—figure supplement 2:

“In immunofluorescence staining experiments, we observed that cortical recruitment of LIN-5 localized dynein to the cortex (Figure 7—figure supplement 2). Notably, the reverse was also seen: we detected LIN-5 at the cortex following the direct recruitment of ePDZ::DHC-1 to PH::LOV, even after knockdown of *gpr-1/2* by RNAi (Figure 6—figure supplement 2, top). The p150(Glued) dynactin subunit DNC-1 was also present at the cell cortex of such embryos, which indicates that at least some PH::LOV-localized dynein complexes contain the dynactin cofactor (Figure 6—figure supplement 2, lower panels). DNC-1^p150^ colocalized with PH::LOV membrane-recruited dynein even in *lin-5(RNAi)* embryos. Thus, the cortical localization of dynein through a direct PH::LOV interaction leads to co-recruitment of LIN-5 and dynactin, but not to significant force generation. It is possible that these complexes adapt an inactive conformation or lack specific subunits of the dynein-dynactin motor complex. In contrast, dynein anchored at the cell cortex through the LIN-5 intermediate generated strong pulling forces. Together, these data indicate that association with membrane-attached LIN-5 is essential for dynein to generate cortical pulling forces.”

In the manuscript, we included only the anti-FLAG staining of DNC-1::mNG::3xFLAG, because the anti-DNC-1 antibody occasionally showed some background staining of the cortex in negative control experiments. Using this reagent in additional experiments, we observed that DNC-1 is also co-recruited with ePDZ::DHC-1 to the cortex in the apparent absence of GPR-1/2 and LIN-5 (New Figure 5—figure supplement 2, lower panels). Although surprising, these results support our conclusion that membrane anchoring of the dynein-dynactin complex through LIN-5 is required for pulling force generation on MT plus ends.

The Kiyomitsu group (Okumura et al., 2018) also observed that dynein needs to be anchored through NuMA for spindle positioning. Moreover, they also observed that, despite the lack of cortical pulling, the p150 subunit is co-recruited when dynein is directly targeted to the cell membrane. A difference from their results is that we observe co-recruitment of LIN-5^NuMA^ as well. The two systems show remarkably different kinetics: using the TULIP system, we observe membrane localization of ePDZ-tagged endogenous proteins within seconds, while in the iLID experiments in HeLa cells accumulation takes place over minutes. It is possible that in this time frame, dynein complexes are recruited to the cortex with adaptors other than NuMA. We have included this point in the Discussion section (see also point 8 below).

2) An interesting analysis of the manuscript is the dissection of the Gα∙GTP-cycle and its regulation by RGS-7 and RIC-8. The authors generically refer to Gα to indicate both GOA-1 and GPA-16. It is known that in Drosophila neuroblasts two Gα subunits synergies to drive spindle orientation (Yoshiura, 2012). To understand whether also in C. elegans the two Gα subunits play distinct roles, it would be interesting to explore the phenotypic effect of ablating of the Gα subunits individually when targeting RGS-7 or RIC-8 to the cortex. Alternatively, the possibility of the two Gα subunits playing distinct roles should be discussed.

Indeed, the GOA-1 and GPA-16 Gα proteins act in a substantially redundant manner, but each protein contributes by itself to cortical pulling forces (Afshar et al., 2004; our unpublished results), which does not necessarily involve identical mechanisms. Several studies have described differences in the contribution of GOA-1 compared to GPA-16 in spindle positioning, including studies by the Gönczy lab (e.g. Afshar et al., 2005) and our group (e.g. Berends et al., 2013). Nevertheless, these distinct roles remain incompletely understood, and cortical RGS-7 and RIC-8 targeting experiments in combination with individual Gα knockdown could indeed reveal specificity of RGS-7 for one of the Gα proteins. It is also possible, but not likely, that an effect from RIC-8 recruitment would be observed in the context of a single Gα, in contrast to the wild-type situation that we have examined.

Although interesting, we feel that including such experiments would take away from the flow of the manuscript and the main focus of this part of the study. We mainly try to address the question whether Gα∙GTP (GOA-1 and/or GPA-16) may contribute to spindle positioning. Therefore, we took the reviewer’s suggestion to expand on possible distinct roles of Gα proteins in the Discussion section.We have now included discussion of *C. elegans* GOA-1 and GPA-16 in a newly added paragraph, and of *Drosophila* Gα_i_ and Gα_o_ in the text preceding and following this paragraph.

3) The authors report that membrane invaginations appear upon GPR-1 ectopic targeting, which is an intriguing observation. Do the authors think that these invaginations are physiological or that they are generated by increased cortical levels of GPR-1? What are the levels of GPR-1 ectopically accumulated at the membrane by the TUPLIP photo-inducible system compared to the levels recruited by Gα? It would be important to discuss whether these invaginations might be a common feature occurring in cellular systems other than C. elegans embryo as a general mechanism of force generators' activation.

We regret that the description of membrane invaginations has not been sufficiently clear. These invaginations can be detected in wild-type embryos and were used by us as a read out to quantify the extent and distribution of spindle pulling forces; thereby providing a method that is independent of the spindle severing assays and complements these assays.

These membrane invaginations were first described by the Hyman and Howard labs (Redemann et al., 2010; cited in the manuscript). These authors make a strong case that the invaginations are physiologic and reflect that the cell membrane is pulled inward at the sites of cortical force generation. The invaginations become much more prominent when the actin-myosin cortex is weakened, and they fully depend on the components of the force generating complex. Therefore, they have been proposed by Redemann et al., to “serve as a tool to localize the sites of force generation at the cortex”. In previous studies, we have followed these invaginations extensively by spinning disk confocal immunofluorescence microscopy (e.g. Berends et al., 2013; Schmidt et al., 2017).

To get this point across more clearly, we have included a more extensive reference to the earlier studies (subsection “Membrane anchoring of GPR-1^Pins/LGN^ in the absence of Gα reconstructs a cortical pulling force generator”):

“Therefore, we used an additional read-out of cortical pulling forces. Cortical pulling events cause inward movements of the plasma membrane, which are visible by spinning disk-confocal fluorescence microscopy as extended invaginations of the plasma membrane (Redemann et al., 2010) (Figure 5G and Video 10, Video 11). These membrane invaginations occur in wild type embryos, depend on microtubules and cortical force generator components, and correlate with the distribution of pulling force generators (Redemann et al., 2010). Therefore, these membrane invaginations most likely reveal the presence and distribution of active individual force generators.“

Moreover, we extended the description and explanation of the change in invaginations (subsection “Membrane anchoring of GPR-1^Pins/LGN^ in the absence of Gα reconstructs a cortical pulling force generator”), which addresses question (4) (see below).

Even though invaginations are detected in wild-type embryos, the reviewer’s question about cortical GPR-1 levels is clearly relevant. Based on immunofluorescence microscopy as well as immunofluorescence staining experiments, the induced levels are well above the normal endogenous level of GPR-1, and also LIN-5 and DHC-1 (see images Figure 5B, Figure 6B, Figure 7B) This also follows from previous quantifications: for eGFP::LIN-5, we recorded peak levels of up to ~1.2-fold enrichment at the cortex-over-cytoplasm in anaphase of the P0 blastomere (Schmidt et al., 2017). Using the same quantification method and immunofluorescent staining of wild type embryos, Park and Rose reported a ~1.2-fold cortex-over-cytoplasm ratio for LIN-5, and ~1.5-fold enrichment for GPR-1/2 (Development 2008). Thus, the induced GPR-1 and LIN-5 cortical levels are much higher than normal (see quantifications in Figure 5C, Figure 7C), but dynein at the cortex does not increase to the same extent (Figure 6—figure supplement 2, Figure 5—figure supplement 1). Because of these differences in cortical protein levels, we only comment on the distribution of pulling forces, not on the absolute amount.

Notably, the cortex-over-cytoplasm of endogenously tagged dynein is closer to 1 in the one-cell embryo (Schmidt et al., 2017). This is well below the ~2 to 3-fold cortex-over-cytoplasm ratio observed for ePDZ::mCherry::DHC-1 in the fully-induced state (See quantification in Figure 6C). Despite this high amount of directly membrane-tethered dynein, pulling forces were not detected.

In the manuscript, we show a quantification of the cortex-over-cytoplasm levels over time following exposure to blue light, for the ePDZ fusions of GPR-1 as well as DHC-1 and LIN-5 (Figure 5C, Figure 6C, Figure 7C). While not representing the normal distribution of endogenous proteins, this illustrates how our phenotypic observations correlate with the induced cortical protein levels.

4) The authors find that Gα knockdown results in a shift in the band of decreased pulling forces that in wild-type zygotes depends on LET-99 for its slightly posteriorized location. While this is a minor issue, it is somewhat puzzling. Have the authors, or others in previous studies, examined LET-99 localization in Gα mutants? If this has been done, it would be worth mentioning in the Discussion section; if not perhaps the authors could look at LET-99 localization and see if it is shifted similarly?

The localization of LET-99 has been previously studied in *goa-1; gpa-16* RNAi embryos by both Bringmann et al., (2007) and Price and Rose (2017).

However, the point we try to make is not that LET-99 becomes displaced, but that the distribution of spindle pulling forces becomes independent of LET-99. In agreement, both cited studies reported a displacement of LET-99 to the posterior, while the dip in invaginations we observe shifts anteriorly (to the middle of the embryo). More importantly, direct recruitment of GPR-1/2 to the cortex overcomes the normal drop in cortical GPR-1/2, which is now stated in the text.

We tried to clarify this issue by re-wording the text in subsection “Membrane anchoring of GPR-1^Pins/LGN^ in the absence of Gα reconstructs a cortical pulling force generator”:

“When plotted along the anterior-posterior axis, the distribution of these invaginations reflected the three described cortical domains: anterior, posterior, and a posterior lateral region at ± 60% embryo length (Rose and Gönczy, 2014) (Figure 5H-left). The posterior lateral band region localizes the LET-99 DEP-domain protein, which antagonizes the localization of GPR-1/2 and thereby pulling force generation (Krueger et al., 2010; Tsou et al., 2003). This explains the absence of invaginations around 60% embryo length (Figure 5H-left).

Cortical GPR-1 recruitment resulted in a total number of 174 (+25% compared to *ph::lov* control) invaginations in the presence, and 122 invaginations (+249% compared to *Gα(RNAi)* embryos) in the absence of Gα (Figure 5—figure supplement 3). Thus, in agreement with our observations in spindle severing assays, GPR-1 recruitment to the membrane induces cortical pulling forces, even in the apparent absence of Gα proteins. The lack of invaginations around 60% embryo length was no longer detected when ePDZ::GPR-1 was recruited to PH::LOV. In agreement, the characteristic dip in cortical GPR-1 localization (e.g.: Figure 1B) was no longer detected after ectopic GPR-1 recruitment (Figure 5B). Thus, as expected, LET-99 does not antagonize the cortical recruitment of ePDZ::GPR-1 by PH::LOV, in contrast to the Gα∙GDP-mediated localization of endogenous GPR-1/2. The pattern of invaginations still showed two peaks and a mild dip at 50% embryo length (Figure 5H). The remaining peak numbers of invaginations likely represent the cortical regions closest to the spindle poles, as these sites contact the highest numbers of astral microtubules. Taken together, Gα is not essential for force generation, but the characteristic distribution of force generating events is likely regulated in part at the Gα protein or Gα–GPR-1/2 protein interaction level.”

5) The authors nicely count spots associated with membrane invaginations as an alternative to pulling forces to quantify the effects of light-induced cortical localization of these regulators. In the images of the spots (Figure 5G), there seem to be spots at the very posterior and anterior poles, and yet the bar graphs show an absence of such foci at the termini. Why are the numbers reduced at the extreme ends, and if they are, why do there seem to be quite a few of them, particularly at the anterior pole, in the lower image of Figure 5G? I can imagine where it is largely a surface area issue, but some clarification on this point might be worthwhile.

We understand this is confusing. Apart from the size, the most important criterion for assigning dots to membrane invaginations is their transient nature. Thereby, we distinguish between membrane invaginations and the extensive “ruffling” of the cortex, which is most noticeable in the anterior during polarity establishment.

To clarify this point, we included in the Materials and methods section and Legend of Figure 5G:

“Plasma membrane invaginations resulting from cortical pulling forces are visible as lines in the DNA plane and dots in the subcortical plane (red arrows). Larger structures are membrane ruffles, which are distinct from the more dynamic invaginations, as can be seen in Video 11. Scale bar: 5 µm.”

6) The authors never present any data in which LIN-5 is knocked down using RNAi. In particular, it would seem like a worthwhile thing to do when documenting the light-induced cortical localization of Dynein (presumably it would be independent of not only GPR-1/2, as shown, but also of LIN-5). Perhaps more importantly, as something of a control, it might be worth showing that the effect of light induced cortical localization of LIN-5 is in fact LIN-5 dependent (loss of pulling forces after blue light exposure in lin-5(RNAi) zygotes). This is a minor concern, given the other controls. But I am curious as to why the authors have not used any lin-5(RNAi) in their analysis at some point, just for the sake of completeness if nothing else.

Indeed, we focused on Gα and *gpr-1/2* knockdown embryos, because these embryos do not deviate from wild-type until the time of mitotic spindle positioning. By contrast, *lin-5* is also required for meiosis and RNAi of *lin-5* causes a failure to expel polar bodies, resulting in the formation of multiple maternal pronuclei (Srinivasan et al., 2003; van der Voet et al., 2009). While the extra chromosomes congress at a single metaphase plate in *lin-5(RNAi)* embryos, it seems cleaner to avoid these additional defects. Nevertheless, we agree with the reviewer that examination of *lin-5(RNAi)* embryos should be included in the manuscript, and we have performed such experiments for the dynein recruitment studies (Figure 6—figure supplement 1, Figure 6—figure supplement 2).

7) While the authors point out how their analysis goes beyond two previous studies that also addressed the role of LIN-5/NuMa in activating cortical dynein pulling forces (Kota et al., 2012; Ségelin et al., 2010), the authors need to address in similar fashion results just published in eLife from Okumura et al., 2018) concerning vertebrate NuMa. The authors should cite this paper and discuss their results in the context of the ones reported by the Kiyomitsu paper.

We agree, and now extensively refer to the Kiyomitsu paper in the Discussion section:

“In two earlier studies, this resulted in the conclusion that the NuMA complex acts as a dynein anchor, but substitution of components or Gα protein removal was not tested (Kotak et al., 2012; Ségalen et al., 2010). However, a very recent study of HeLa cells followed an optogenetic strategy similar to ours, and also observed that a membrane anchor cannot replace the entire Gα–LGN–NuMA complex (Okumura et al., 2018). As in *C. elegans* embryos, a CAAX membrane anchor could substitute for Gα–LGN in Hela cells, while dynein needed to be anchored through NuMA in order to generate spindle positioning forces (Okumura et al., 2018). Thus, observations in two different systems indicate that dynein activation at microtubule plus ends requires the LIN-5/NuMA adaptor protein, similar to the requirement for an activating dynein adaptor in cargo transport (McKenney et al., 2014; Schlager et al., 2014; Zhang et al., 2017).**”**

“……In the optogenetic experiments in Hela cells, dynein also localized p150 at the membrane, however in contrast to our findings, NuMA was not observed to be co-recruited in this study (Okumura et al., 2018). While the reason for this discrepancy is currently unknown, it might be related to the remarkably different kinetics of the two systems. Using the TULIP system, we observed membrane localization of ePDZ-tagged endogenous proteins within seconds, whereas in the iLID experiments in HeLa cells accumulation took place over multiple minutes. It is possible that in the longer time frame, cytoplasmic dynein complexes are recruited to the cortex with adaptors other than NuMA.”

8) Unless I missed it, the authors do not describe in the text, figure legend, or methods exactly how they did the localized light-induced increases in cortical LIN-5. Some mention of the control of blue light illumination used for restricted localization should be provided in the Materials and methods section at least.

This was indeed included, and most likely missed because it was described Materials and methods section titled ‘Dark state spinning disc microscopy’. We renamed this section ‘Dark state experiments and local recruitment of ePDZ-tagged proteins to membrane LOV’ for clarity. The hardware and software used for these experiments are mentioned in subsection ‘Spinning disc microscopy’.

The sensitivity of the ePDZ-LOV interaction and scattering of light varies from system to system, depending on protein concentrations and physical properties of the cell under investigation. The area of activation and laser power appropriate for local control of ePDZ-LOV activation will therefore have to be determined by experience of the experimentalist. For this purpose, we used a control strain expressing diffuse ePDZ::mCherry and PH::eGFP::LOV to calibrate the laser power and area of activation that enable local control of the ePDZ-LOV interaction. The laser powers we used ranged from 1-10%, depending on the strength of the laser (we have an in-house built microscopy set-up for which the lasers have to be re-aligned manually from time to time, after which laser power is stronger).